**Resource**

**EMBO** *reports*

# Transcript-specific induction of stop codon readthrough using a CRISPR-dCas13 system

Lekha E Manjunath[1], Anumeha Singh [1,2], Sangeetha Devi Kumar[1], Kirtana Vasu[1], Debaleena Kar[1], Karthi Sellamuthu[1,3] & Sandeep M Eswarappa [1 ✉]

## Abstract

**Stop codon readthrough (SCR) is the process where translation continues beyond a stop codon on an mRNA. Here, we describe a strategy to enhance or induce SCR in a transcript-selective manner using a CRISPR-dCas13 system. Using specific guide RNAs, we target dCas13 to the region downstream of canonical stop codons of mammalian *AGO1* and *VEGFA* mRNAs, known to exhibit natural SCR. Readthrough assays reveal enhanced SCR of these mRNAs (both exogenous and endogenous) caused by the dCas13-gRNA complexes. This effect is associated with ribosomal pausing, which has been reported for several SCR events. Our data show that CRISPR-dCas13 can also induce SCR across premature termination codons (PTCs) in the mRNAs of green fluorescent protein and TP53. We demonstrate the utility of this strategy in the induction of readthrough across the thalassemia-causing PTC in *HBB* mRNA and hereditary spherocytosis-causing PTC in *SPTA1* mRNA. Thus, CRISPR-dCas13 can be programmed to enhance or induce SCR in a transcript-selective and stop codon-specific manner.**

**Keywords** Cas13; CRISPR; Nonsense Mutation; Stop Codon; Translational Readthrough
**Subject Categories** Methods & Resources; Translation & Protein Quality

## Introduction

In certain transcripts, ribosomes fail to terminate translation at the stop codon and continue till the next in-frame stop codon, resulting in longer protein isoforms with extended C-termini. This phenomenon called stop codon readthrough (SCR) is mediated by the recoding of stop codons by near-cognate tRNAs or suppressor tRNAs (Schueren and Thoms, 2016). Readthrough isoforms can differ from the canonical isoforms in their function, localization and/or stability (Eswarappa et al, 2014; Manjunath et al, 2020; Schueren et al, 2014; Singh et al, 2019). Hence, target-specific manipulation of SCR can have potential applications. For example, SCR of *VEGFA* results in reduced angiogenesis, which can

be beneficial in the treatment of cancerous tumors and retino-pathies (Eswarappa et al, 2014). Furthermore, transcript-specific induction of SCR will have therapeutic benefits in genetic diseases caused by nonsense mutations (e.g., β-thalassemia) that result in premature stop codons (or termination codons, PTCs) in the coding sequences (Keeling et al, 2014).

Ataluren, certain macrolide and aminoglycoside antibiotics, and 2,6-diaminopurine are some of the molecules that can induce SCR across premature termination codons (PTCs) (Howard et al, 1996; Trzaska et al, 2020; Welch et al, 2007; Zilberberg et al, 2010). Engineered suppressor tRNAs is another strategy under development for the induction of SCR (Lueck et al, 2019). However, all these strategies lack specificity. They can potentially target any stop codon in any transcript, and result in undesired effects. Therefore, there is a need for a strategy that can induce SCR in a transcript-specific manner.

hnRNPA2/B1, an RNA-binding protein, binds downstream of the stop codon (9 nucleotides downstream) and positively regulates the SCR of *VEGFA* (Eswarappa et al, 2014; Nico et al, 2021). Similarly, let-7a micro-RNA binds downstream of the stop codon of *AGO1* mRNA and enhances its SCR (Singh et al, 2019). Even exogenous antisense oligonucleotides that target the region downstream of the PTC in *HBB* mRNA can induce SCR (Kar et al, 2020). Furthermore, RNA pseudoknot structure formed downstream of the stop codon present between the *gag* and *pol* region of murine leukemia virus induces SCR (Houck-Loomis et al, 2011). These observations indicate that transient molecular obstacles downstream of the stop codon can enhance or induce SCR (Manjunath et al, 2022). Therefore, targeting of an RNA-binding protein specifically downstream of a stop codon can potentially serve this purpose. Here, we have used CRISPR-dCas13 system to achieve this.

RNA targeting Cas13 (formerly C2c2) belongs to the Class 2 Type VI CRISPR-Cas system. This CRISPR-associated enzyme targets RNA in a sequence-specific manner with the help of a guide RNA (Watanabe et al, 2019). Because of this specificity, multiple applications such as RNA editing, DNA/RNA detection, RNA knockdown and dynamic imaging of RNAs have been developed (Abudayyeh et al, 2017; Cox et al, 2017; Gootenberg et al, 2017; Yang et al, 2019). This system has also been used in the diagnosis and treatment of SARS-CoV-2 infections in a rodent model (Blanchard et al, 2021; Freije et al, 2019; Patchsung et al, 2020). Here, we have applied CRISPR-dCas13 (catalytically inactive or

[1]Department of Biochemistry, Indian Institute of Science, Bengaluru, Karnataka 560012, India. [2]Present address: Northwestern University Feinberg School of Medicine, Chicago, IL, USA. [3]Present address: University of Texas Medical Branch, Galveston, TX, USA. ✉E-mail: sandeep@iisc.ac.in

dead Cas13) system to target mRNAs proximally downstream of the stop codon to enhance or induce SCR in a transcript-specific manner.

# Results

Since Cas13 can target a specific region of an mRNA with the help of a guide RNA (gRNA), we hypothesized that this system can be used to create a transient molecular obstacle for ribosomes near the stop codon, and thereby, enhance or induce SCR. We tested this hypothesis using the catalytically inactive variant of this enzyme—dCas13, which can bind a specific region of an mRNA without cleaving it (Abudayyeh et al, 2017). The gRNAs were designed such that they target mRNAs proximally downstream (4–40 nucleotides) of the stop codon.

## Targeted enhancement of SCR by dCas13

We first tested our hypothesis in *AGO1* mRNA, which encodes Argonaute 1 (Ago1) protein. Ago1 is important for micro-RNA-mediated repression of gene expression (Meister, 2013). *AGO1* has been shown to undergo SCR across its canonical stop codon resulting in a longer isoform called Ago1x (Ghosh et al, 2020; Singh et al, 2019). Unlike Ago1 (the canonical isoform), Ago1x cannot repress the expression of target transcripts and serves as an inhibitor of the micro-RNA pathway when its expression is increased (Singh et al, 2019). In cancer cells, Ago1x inhibits dsRNA-induced interferon signaling and promotes cell proliferation (Ghosh et al, 2020).

We designed a gRNA such that it guides dCas13 (from *Leptotrichia wadei*) to the region downstream of the canonical stop codon of *AGO1* mRNA (blue region in Fig. 1A). A gRNA that does not target any transcript in mammalian cells (nontargeting) was used as control in all assays (Abudayyeh et al, 2017). Expression of dCas13 in transfected cells was confirmed by RT-PCR (Fig. EV1A). Immunoprecipitation of FLAG-tagged dCas13 followed by qRT-PCR confirmed the interaction of dCas13 with the *AGO1* mRNA in the presence of the *AGO1*-targeting gRNA (Fig. EV1B).

To test if dCas13 can enhance the SCR of *AGO1*, we first performed a western blotting-based readthrough assay (Singh et al, 2019). Part of the *AGO1* coding sequence (696 nucleotides of the 3′-end) along with its canonical stop codon and the proximal part of its 3′UTR (99 nucleotides) were cloned upstream of and in-frame with FLAG-HA tag. Expression of FLAG-HA-tagged protein is expected only if there is SCR across the canonical stop codon of *AGO1* (schematic in Fig. 1B). We transfected this construct in HEK293 cells to investigate the SCR. In agreement with our previous study, we observed SCR of *AGO1* across its canonical stop codon in the presence of the proximal part of its 3′UTR as indicated by the band corresponding to FLAG-HA-tagged protein (~35 kDa) in the western blot (Singh et al, 2019). Importantly, there was a significant enhancement in the expression of FLAG-HA-tagged readthrough product in cells expressing dCas13 and *AGO1*-targeting gRNA compared to those expressing a nontargeting gRNA. Neither dCas13 nor *AGO1*-targeting gRNA alone could cause this increased expression of the readthrough product (Figs. 1C and EV1C,EV1D). This assay suggests that dCas13 along with *AGO1*-targeting gRNA can enhance the SCR across the canonical stop codon of *AGO1*.

We then performed a luminescence-based SCR assay to confirm these results. This assay involves constructs similar to the one described above, but with the coding sequence of firefly luciferase (FLuc) cloned downstream to and in-frame with *AGO1* (Singh et al, 2019). FLuc activity, i.e., luminescence, is expected only if there is SCR across the stop codon of *AGO1*. This luciferase construct was in vitro transcribed, and the resultant mRNA was in vitro translated using rabbit reticulocyte lysate (RRL). As reported previously, we observed SCR of *AGO1* across its canonical stop codon in the presence of the proximal part of its 3′UTR as indicated by the increased luminescence (Fig. 1E, left). We then performed the RRL-mediated in vitro translation in the presence of extracts from HEK293 cells expressing dCas13 and the *AGO1*-targeting gRNA. We observed a significant increase in the FLuc activity (i.e., SCR) in the presence of both dCas13 and *AGO1*-targeting gRNA. qRT-PCR analysis showed that there was no change in FLuc mRNA levels in these conditions (Fig. 1E, right, F).

Similar results were observed when the luminescence-based SCR assay was performed in HEK293 cells. There was a modest enhancement in the SCR in cells expressing dCas13 and *AGO1*-targeting gRNA compared to those expressing a nontargeting gRNA (3rd and 4th bar Fig. EV1E). Importantly, the combination of *AGO1*-targeting gRNA and dCas13 failed to alter the activity of the luciferase construct that lacked the proximal part of *AGO1*-3′UTR, which is the target region of the gRNA (1st and 2nd bars in Fig. EV1E). This shows that the gRNA and dCas13 work in a transcript-specific manner to enhance SCR. Furthermore, this combination did not show detectable change in the normal translation of the target mRNA as indicated by the luminescence in cells transfected with *AGO1*-3′UTR-FLuc construct without any stop codon between them (Fig. EV1F). qRT-PCR analyses revealed no change in the levels of the target RNA, i.e, *AGO1*-luciferase mRNA, in the presence of dCas13 and the gRNAs (Fig. EV1E,F). Since luminescence is an indicator of SCR, this assay suggests that dCas13 along with *AGO1*-targeting gRNA can enhance the SCR across the canonical stop codon of *AGO1*.

Since multiple plasmid constructs are involved in these assays, alterations in transfection efficiency can potentially bias our results. To rule this out, we quantitated co-transfection efficiency by flow cytometry using mCherry-expressing plasmid as a neutral reporter (i.e., not a target of the gRNA). Neither gRNA (*AGO1*-targeting) expression nor dCas13 expression, individually or together, altered the transfection efficiency of mCherry-expressing plasmid (Fig. EV2A). Furthermore, qRT-PCR analysis of reporter mRNA levels did not change in the conditions we tested (Figs. 1D and EV1E).

The reporter-based assays can also be potentially biased by cryptic promoter activity, alternative splicing, and reinitiation due to leaky scanning near the reporter. However, these are unlikely as the cloned proximal 3′UTR of *AGO1* (target of gRNA-dCas13) is just 99 nucleotides, and it does not have any ATG in-frame with the reporter. Furthermore, the western blotting-based SCR assay described in Fig. 1C rules out most of these possibilities, which would have resulted in shorter products than what we have observed. Also, the in vitro translation assay described in Fig. 1E cannot be biased by cryptic promoter activity or alternative splicing. In addition, we sequenced the cDNAs of the mRNAs isolated from the cells transfected with reporter constructs and plasmids expressing dCas13 and gRNAs to rule out alternative splicing events (Fig. EV2B,EV2C; Appendix Figs. S1 and S2).

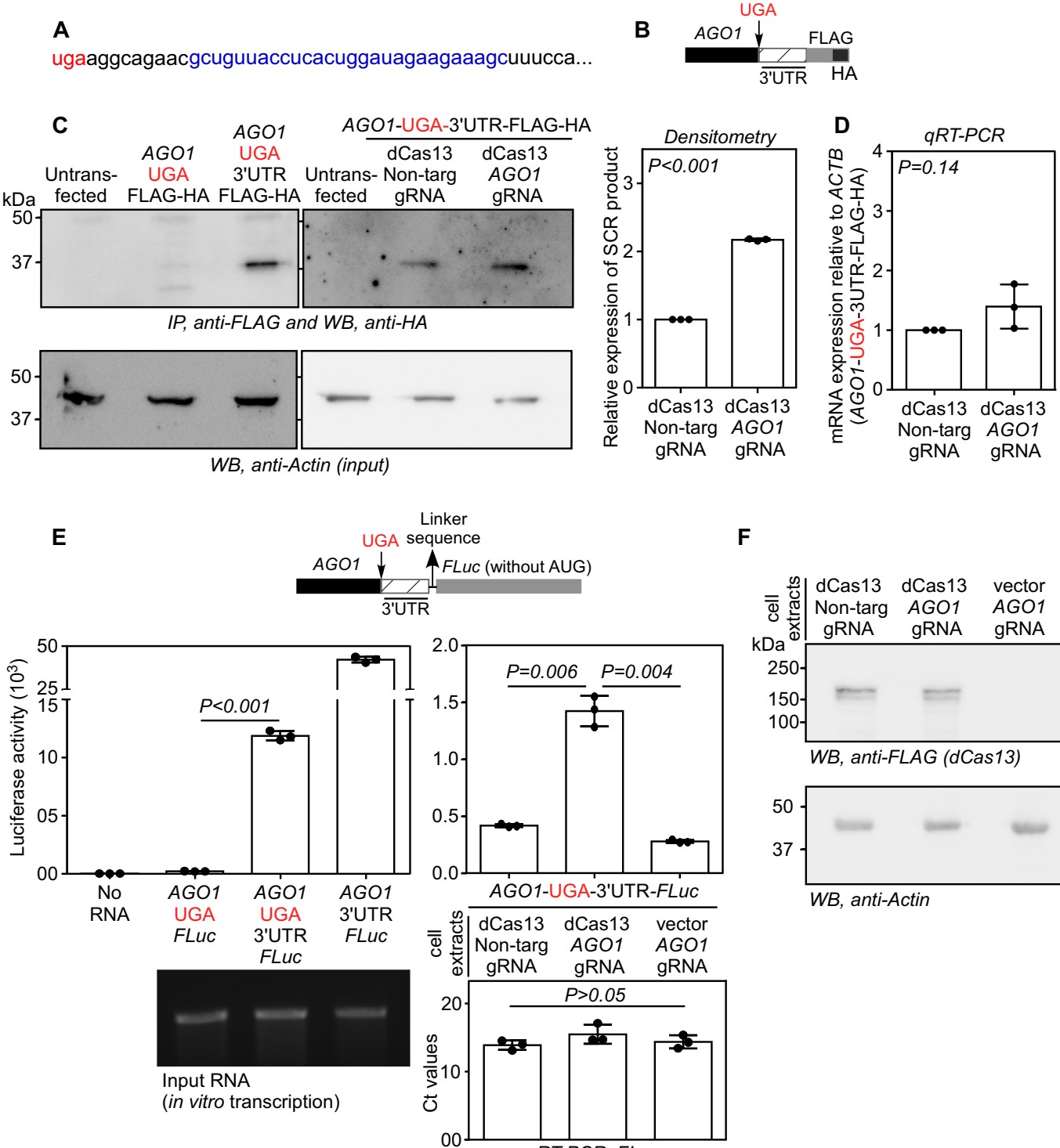

**A**

ugaaggcagaacgcuguuaccucacuggauagaagaaagcuuucca...

**B**
AGO1 — UGA — 3'UTR — FLAG — HA

**C**

IP, anti-FLAG and WB, anti-HA

WB, anti-Actin (input)

*Densitometry* — P<0.001

**D** *qRT-PCR* — P=0.14

**E**
AGO1 — UGA — Linker sequence — FLuc (without AUG) — 3'UTR

Luciferase activity (10³)
P<0.001

Input RNA (*in vitro* transcription)

AGO1-UGA-3'UTR-FLuc
P=0.006  P=0.004

qRT-PCR, FLuc
P>0.05

**F**

WB, anti-FLAG (dCas13)

WB, anti-Actin

## dCas13 enhances the stop codon readthrough of endogenous *AGO1*

We next tested if dCas13 can increase the SCR of endogenous *AGO1*, which results in the longer isoform Ago1x. HEK293 cells transfected with plasmids expressing *AGO1*-targeting gRNA and dCas13 showed about 2.5-fold increase in the expression of Ago1x protein compared to those transfected with a nontargeting gRNA. *AGO1*-targeting gRNA alone was unable to cause this increase. The higher molecular weight band observed with anti-Ago1x antibody might represent post-translationally modified isoform as reported previously (Leung et al, 2011) (Fig. 2A). There was no detectable change in the total Ago1 protein and *AGO1* mRNA levels under the same conditions (Fig. 2A,B). The gRNA-mediated enhancement of

**Figure 1.  *AGO1*-targeting dCas13 enhances SCR of *AGO1* mRNA.**

(A) The sequence of the proximal 3′UTR of *AGO1*. The canonical stop codon (red) and the gRNA targeting region (blue) are highlighted. (B) Schematic showing the test construct used in the western blotting-based SCR assay. The FLAG-HA tag was cloned downstream of, and in-frame with the partial coding sequence of *AGO1* (696 nucleotides of the 3′-end) along with its proximal 3′UTR (99 nucleotides) such that FLAG-HA-tagged protein is expressed only after SCR. Another construct without the proximal 3′UTR was used as a negative control. (C) Western blotting-based SCR assay. The indicated plasmid constructs were transfected in HEK293 cells. After 48 h, they were subjected to immunoprecipitation followed by western blotting to detect FLAG-HA-tagged SCR product. The bar graph shows the expression levels of SCR product quantified by densitometry. (D) qRT-PCR analysis of the expression of FLAG-HA-tagged reporter construct relative to *ACTB*. (E) Luminescence-based in vitro translation assay to quantify SCR. The schematic shows the test construct, which is similar to the one shown in (B), except that the coding sequence of firefly luciferase (FLuc) is present in place of FLAG-HA tag. Luciferase activity (luminescence) is expected only if there is SCR. The constructs were in vitro transcribed and the obtained RNAs were in vitro translated using rabbit reticulocyte lysate. The graph on the left shows luminescence from indicated constructs. The graph on the right shows luminescence values in the presence of HEK293 cell extracts expressing dCas13 and indicated gRNAs. Ct values obtained by qRT-PCR shows the levels of FLuc mRNA after incubation with HEK293 cells extracts for 2 h. (F) Western blot showing the expression of dCas13 in the HEK293 cell extracts. Data information: (C–E) Graphs show mean ± SD, $N = 3$ experiments. Two-sided Student's *t* test was used to calculate the *P* values. Source data are available online for this figure.

Ago1x expression was observed using dCas13 from another bacterium, *Porphyromonas gulae* (dPguCas13) also (Fig. 2C). These results show that the designed *AGO1*-targeting gRNA along with dCas13 increases the expression of Ago1x, the readthrough product of *AGO1*. Importantly, this action does not affect the canonical translation of *AGO1* mRNA or its levels. This implies that the increased Ago1x expression is because of increased SCR across the canonical stop codon of *AGO1* mRNA. These observations are consistent with the reporter-based readthrough assays described above (Fig. 1).

The *AGO1*-targeting gRNA described above reduced the expression of Ago1x when transfected along with Cas13, which is catalytically active (Fig. EV3A). This result along with the results shown in Fig. EV1B and Fig. 2 shows that the *AGO1*-targeting gRNA can guide Cas13 and dCas13 to its target mRNA, i.e., *AGO1*. Furthermore, targeting the same *AGO1*-3′UTR region (Fig. 1A) on the genomic DNA using CRISPR–Cas9 system knocked down the expression of Ago1x (Fig. EV3B). Complete knockout of Ago1x could not be achieved, possibly because of lethality consistent with a previous report (Ghosh et al, 2020). This result shows that the band we observed in western blotting assay indeed represents the SCR product of *AGO1*, i.e., Ago1x.

Since overexpression of Ago1x can enhance global translation (Singh et al, 2019), we investigated the effect of dCas13-mediated induction of *AGO1* SCR on global translation. For this, we employed RiboPuromycylation assay (Bastide et al, 2018). In this assay, puromycin is incorporated into nascent peptides during translation as it mimics tyrosyl-tRNA. This can be detected by western blotting using anti-puromycin antibody. Knocking down of *AGO1* using Cas13 and the *AGO1*-targeting gRNA in HEK293 cells showed higher puromycin incorporation compared to control cells demonstrating the expected increase in global translation due to the reduction in the Ago1 levels (Fig. EV3C). Similar effect was observed in cells overexpressing Ago1x, which is a negative regulator of micro-RNA function as shown previously (Fig. EV3D,EV3E) (Singh et al, 2019). These two results showed that RiboPuromycylation assay can be used to investigate the effect of induction of *AGO1* SCR on global translation. We performed this assay in cells transfected with dCas13 along with *AGO1*-targeting gRNA. We observed an enhanced puromycin signal, indicating increased global translation, in these cells compared to control cells (Fig. 3A). This observation is consistent with increased Ago1x expression resulting from increased SCR of *AGO1*.

To confirm these findings, we used a quantitative fluorescence-based assay. This assay involves treating cells with L-homopropargylglycine (HPG, an analog of methionine), which contains an alkyne moiety. HPG is incorporated into proteins during translation. Addition of the Alexa Fluor™ 488 azide results in 'click' reaction between the azide (Alexa Fluor) and the alkyne (HPG). Because of this reaction, the newly synthesized proteins can be detected by fluorescence and quantified. We applied this method to investigate the effect of *AGO1* SCR induction on global translation. Cells expressing dCas13 and *AGO1*-targeting gRNA showed significantly higher HPG fluorescence intensity by flow cytometry compared to cells expressing nontargeting gRNA. This observation is consistent with the induction of *AGO1* SCR by dCas13 and *AGO1*-targeting gRNA (Fig. 3B). Together, these results show that the induction of SCR of *AGO1* by dCas13 results in a functional product.

The SCR of *AGO1* is positively regulated by let-7a micro-RNA. Both let-7a micro-RNA and the dCas13-gRNA we have used in this study target the same region in the 3′UTR of *AGO1* mRNA. Our results show that, like let-7a, exogenous dCas13-gRNA can induce SCR of *AGO1*. Since the gRNA is 28/30 nucleotides long with 100% complementarity with *AGO1* mRNA, dCas13-gRNA is likely to be a better inducer of SCR than let-7a micro-RNA, which has a 11 nucleotide stretch of partial complementarity with the *AGO1* mRNA (Singh et al, 2019).

## dCas13 causes ribosomal pausing on target mRNA

SCR and ribosomal frameshifting induced by *cis*- and *trans*-acting factors and by small molecules are associated with ribosomal pausing at the stop codon (Annibaldis et al, 2020; Bao et al, 2020; Dever et al, 2018; Lashkevich et al, 2020; Lawson et al, 2021; Seidman et al, 2011; Sharma et al, 2021). Therefore, we tested if dCas13-gRNA complex can cause ribosome pausing at the target mRNA. We adopted a previously established dual-fluorescence-based ribosomal stalling assay (Juszkiewicz and Hegde, 2017). In this assay, GFP and mCherry are expressed from a single mRNA. P2A sites inserted between them (schematic in Fig. 4A) ensure that the two proteins are independent after translation. Ribosomal pausing or stalling between them would result in reduced mCherry signal relative to the GFP signal. As reported earlier, the presence of a strong ribosomal stalling sequence (20 repeats of lysine codon, AAA) resulted in a significant reduction in the ratio of the mean fluorescence intensity of mCherry to that of GFP in transfected cells (Juszkiewicz and Hegde, 2017). To investigate the effect of dCas13 binding, we introduced the proximal 3′UTR of *AGO1* mRNA (99 nucleotides, target of gRNA-dCas13) between the coding sequences

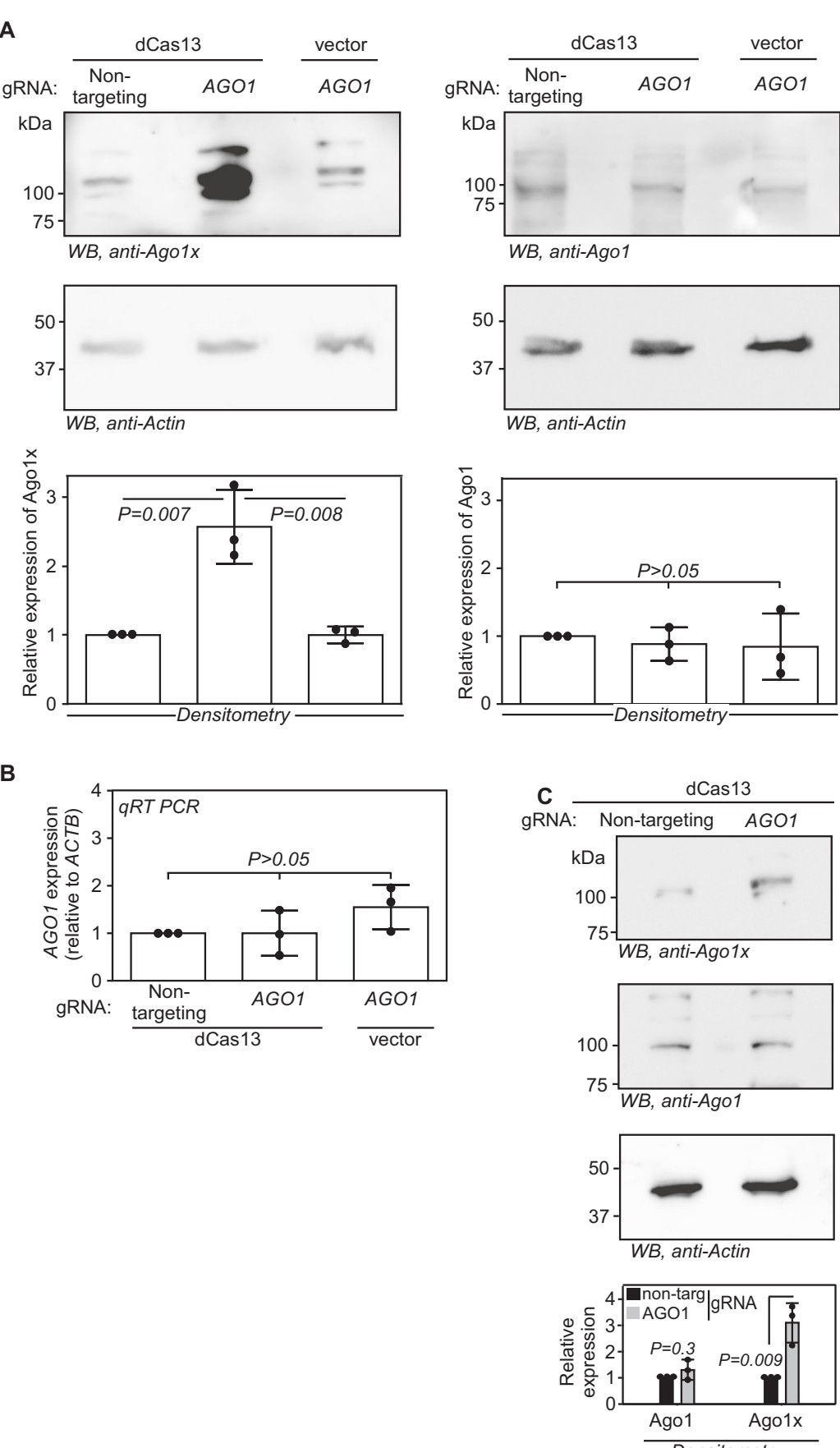

**Figure 2. *AGO1*-3′UTR-targeting dCas13 enhances the expression of Ago1x, the SCR product of endogenous *AGO1* mRNA.**

(**A**) Western blot showing the expression of Ago1x and Ago1 in HEK293 cells expressing dCas13 (from *L. wadei*) and *AGO1*-targeting gRNA. Graphs show quantification of expression by densitometry. (**B**) qRT-PCR result showing the expression of *AGO1* mRNA in the same cells. (**C**) Western blot showing the expression of Ago1x and Ago1 in HEK293 cells expressing dCas13 (from *P. gulae*) and *AGO1*-targeting gRNA. Data information: Graphs in (**A–C**) represent Mean ± SD, $N = 3$ experiments. Two-sided Student's *t* test was used to calculate the *P* values. Source data are available online for this figure.

of GFP and mCherry (schematic in Fig. 4A). The construct was transfected in HeLa cells along with plasmids expressing dCas13 and *AGO1*-targeting gRNA. The fluorescence intensities were quantified by flow cytometry 24 h after transfection. We observed a modest reduction in the ratio of the mean fluorescence intensity of mCherry to that of GFP in cells expressing dCas13 and *AGO1*-targeting gRNA (Fig. 4A). This observation suggests ribosomal pausing caused by dCas13 binding. There was no significant change in the levels of GFP-mCherry mRNA in the presence of *AGO1*-targeting gRNA (Fig. 4B).

To further investigate ribosomal pausing, we used a luciferase-based assay. In this assay, the coding sequence of firefly luciferase (FLuc) was tagged with the proximal 3′UTR of *AGO1* (99 nucleotides, target of *AGO1* gRNA-dCas13) at its 3′ end. Ribosomal pausing during termination will cause a delay in the release of the newly synthesized luciferase protein and therefore its activity (i.e., luminescence), which can be quantified (Lashkevich et al, 2020). We performed in vitro transcription of the luciferase constructs, and the obtained mRNAs were used for in vitro translation using rabbit reticulocyte lysate. We first tested this assay using a known SCR-inducing sequence from the Venezuelan equine encephalitis virus (VEEV) (Lashkevich et al, 2020). This sequence showed a delay in the appearance of luminescence compared to a non-specific sequence, which is consistent with ribosomal pausing. Similar results were obtained in the presence of the proximal 3′ UTR of *AGO1* (99 nucleotides), which is also an SCR-inducing sequence (Fig. 4C).

We then performed this assay in the presence of extracts from HEK293 cells expressing dCas13 and *AGO1*-targeting gRNA or nontargeting gRNA. We observed a delay in the appearance of the luminescence in the presence of *AGO1*-targeting gRNA compared to nontargeting gRNA suggesting ribosomal pausing. Importantly, this delay was not observed when FLuc was tagged with a non-specific sequence of same length (Fig. 4D). The cell extracts did not affect the target mRNA levels (Fig. 4E). Together, these two assays show that dCas13-gRNA complex, like other SCR-inducing *trans*-acting factors, causes ribosomal pausing. This pausing might provide sufficient opportunity for the near-cognate tRNAs to recode the stop codon enabling SCR. Similar to our observations, CRISPR-Cas12 system has been shown to cause ribosomal pausing and induce ribosomal frameshifting (Huang et al, 2023).

## dCas13 enhances the stop codon readthrough of *VEGFA*

Next, we tested the ability of dCas13 to induce SCR in another mRNA. We chose *VEGFA* mRNA, which encodes a secretory pro-angiogenic protein vascular endothelial growth factor-A (VEGFA). The SCR of *VEGFA* results in a longer isoform termed VEGF-Ax with a unique C-terminus, which prevents its binding to Neuropilin 1, an important co-receptor in VEGFA signaling. Because of this, VEGF-Ax shows anti-angiogenic or weak angiogenic properties (Eswarappa et al, 2014; Xin et al, 2016).

We designed a gRNA to target the *VEGFA* mRNA downstream of the canonical stop codon (Fig. 5A). This gRNA was expressed in HEK293 cells along with dCas13, and the secreted VEGF-Ax was detected in the conditioned medium by western blotting. The level of secreted endogenous VEGF-Ax was increased in cells transfected with both dCas13 and *VEGFA*-targeting gRNA as compared to those expressing nontargeting gRNA. However, there was no change in the levels of *VEGFA* mRNA under the same conditions (Figs. 5B,C). These results show that dCas13, guided by the specific gRNA, enhances the expression of VEGF-Ax, the SCR product of *VEGFA*. The same *VEGFA*-targeting gRNA caused knockdown of VEGF-Ax when expressed along with Cas13 (catalytically active form) in these cells confirming the ability of this gRNA to target *VEGFA* mRNA (Fig. 5D). Using CRISPR–Cas9 system, we generated VEGF-Ax knockout cells by targeting the proximal 3′ UTR of *VEGFA* (Fig. 5E; Appendix Fig. S4), the region responsible for SCR. These cells showed complete absence of VEGF-Ax confirming that the ~20 kDa band we observe in western blots is the product of SCR of *VEGFA* (Fig. 5E).

Next, luminescence-based readthrough assay was performed to further investigate the ability of dCas13 to enhance the SCR of *VEGFA*. *VEGFA* coding sequence and the proximal part of its 3′UTR (63 nucleotides) were cloned upstream of and in-frame with firefly luciferase coding sequence (schematic in Fig. 5F) (Eswarappa et al, 2014). This construct was expressed in HEK293 cells along with dCas13 and the gRNA. SCR across the canonical stop codon results in luciferase expression which can be quantified as luminescence. We observed increased relative luminescence in cells expressing *VEGFA*-targeting gRNA compared to those expressing nontargeting gRNA (2nd and 3rd bars in Fig. 5F) showing enhancement of SCR. There was no change in the luciferase mRNA level in any of these conditions (Fig. 5F). Furthermore, the translation of *VEGFA*-3′UTR-FLuc construct, which did not have any stop codon in between, was unaltered by the expression of *VEGFA*-targeting gRNA and dCas13 (Fig. 5G). Together, these results show that CRISPR-dCas13 system can be used to enhance SCR across the canonical stop codon of *VEGFA*, without affecting the cellular levels of *VEGFA* mRNA.

Because VEGF-Ax is anti-angiogenic (Eswarappa et al, 2014) or weakly angiogenic (Xin et al, 2016), compared to the canonical isoform VEGF-A, SCR of *VEGFA* mRNA will result in a net anti-angiogenic effect. Therefore, enhancement of SCR in *VEGFA* will be useful to treat diseases with excessive and/or abnormal angiogenesis such as cancer and retinopathies.

We also tested the ability of gRNA-dCas13 system to induce SCR in *MTCH2*, another mRNA known to exhibit SCR (Eswarappa et al, 2014; Manjunath et al, 2020). dCas13 coupled with *MTCH2*-targeting gRNA enhanced SCR across the canonical stop codon of *MTCH2* in a transcript-selective manner without altering the levels of its mRNA or its canonical translation in luminescence-based SCR assays (Fig. EV4). Together, these results show that CRISPR-dCas13 system can be used to enhance SCR without affecting the target mRNA levels.

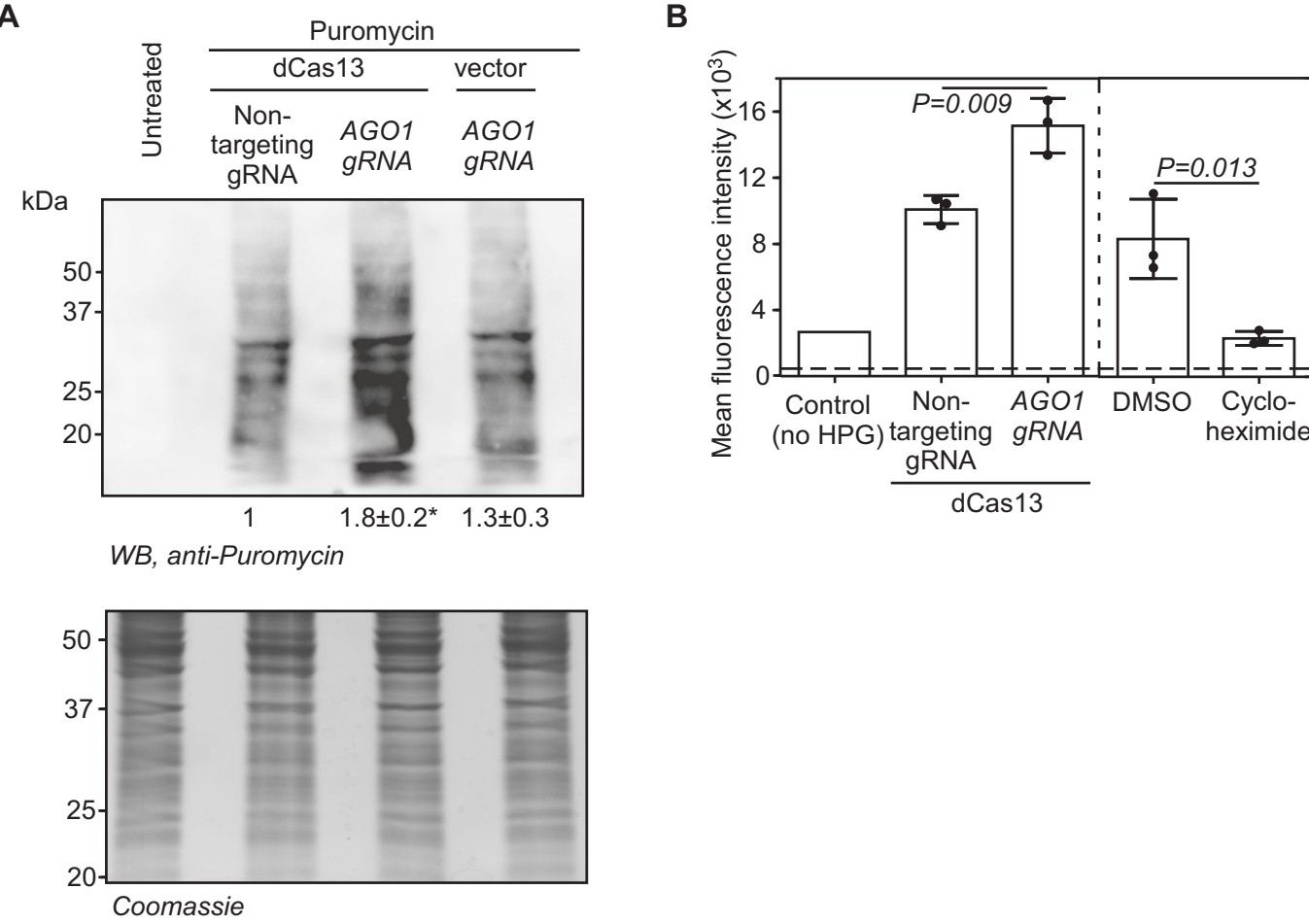

**Figure 3. *AGO1*-3′UTR-targeting dCas13 enhances global translation.**

(A) RiboPuromycylation assay was performed in HEK293 cells expressing dCas13 and *AGO1*-targeting gRNA as described in "Methods". Numbers below indicate the densitometry values (normalized to Coomassie staining) (mean ± SE, $N = 3$ experiments). *$P = 0.02$, Two-sided Student's *t* test. (B) Fluorescence-based protein synthesis assay. HEK293 cells transfected with indicated constructs expressing dCas13 and gRNAs were treated with L-homopropargylglycine (HPG) and Alexa Fluor 488 azide. Fluorescence was quantified by flow cytometry. The graph represents mean ± SD, $N = 3$ experiments. Dotted horizontal line represents the background fluorescence of untreated cells. Cycloheximide, a known translation inhibitor, was used as assay control. Two-sided Student's *t* test was used to calculate the *P* values. Source data are available online for this figure.

## dCas13 can be programmed to induce translational readthrough across premature termination (stop) codons (PTC)

Nonsense mutations resulting in PTCs are responsible for about 11% of genetic diseases (Keeling et al, 2014). We investigated if dCas13 can be used to induce translational readthrough across PTCs. For this, we used a GFP construct with a PTC at 57th codon (TGG to TGA or W57* mutation) designed to test SCR-inducing ability of compounds (Halvey et al, 2012). We designed a gRNA that targets the region downstream of the PTC. We observed full-length GFP, despite a PTC, in HEK293 cells expressing dCas13 along with the GFP-targeting gRNA, but not in cells expressing dCas13 along with nontargeting gRNA, or only GFP-targeting gRNA (without dCas13). This result demonstrates the ability of dCas13-gRNA complex to induce readthrough across the PTC (Fig. 6A).

β-thalassemia is a condition caused by nonsense mutations in *HBB* gene (encodes β-globin protein). This condition is characterized by reduced hemoglobin levels. Worldwide, 80 to 90 million people carry the thalassemia allele in *HBB* gene (Origa, 2017). Any strategy that can induce SCR across the PTCs can provide therapeutic benefit to β-thalassemia patients with nonsense mutations. We tested the ability of CRISPR-dCas13 system to induce SCR in β-thalassemia context. We used an *HBB* construct with a PTC at its 16th codon (TGG to TAG or W16* mutation) as described previously (Fig. 6B) (Kar et al, 2020). This nonsense mutation (w16*) is frequently found in β-thalassemia patients (Colah et al, 2009; Kazazian et al, 1984). The coding sequence of GFP was cloned in-frame with and downstream of *HBB*<sup>w16*</sup> (schematic in Fig. 6B) such that readthrough across the thalassemia-causing PTC will result in full-length β-globin protein tagged with GFP. This construct along with plasmids expressing dCas13 and *HBB*-targeting gRNA (targets the region downstream

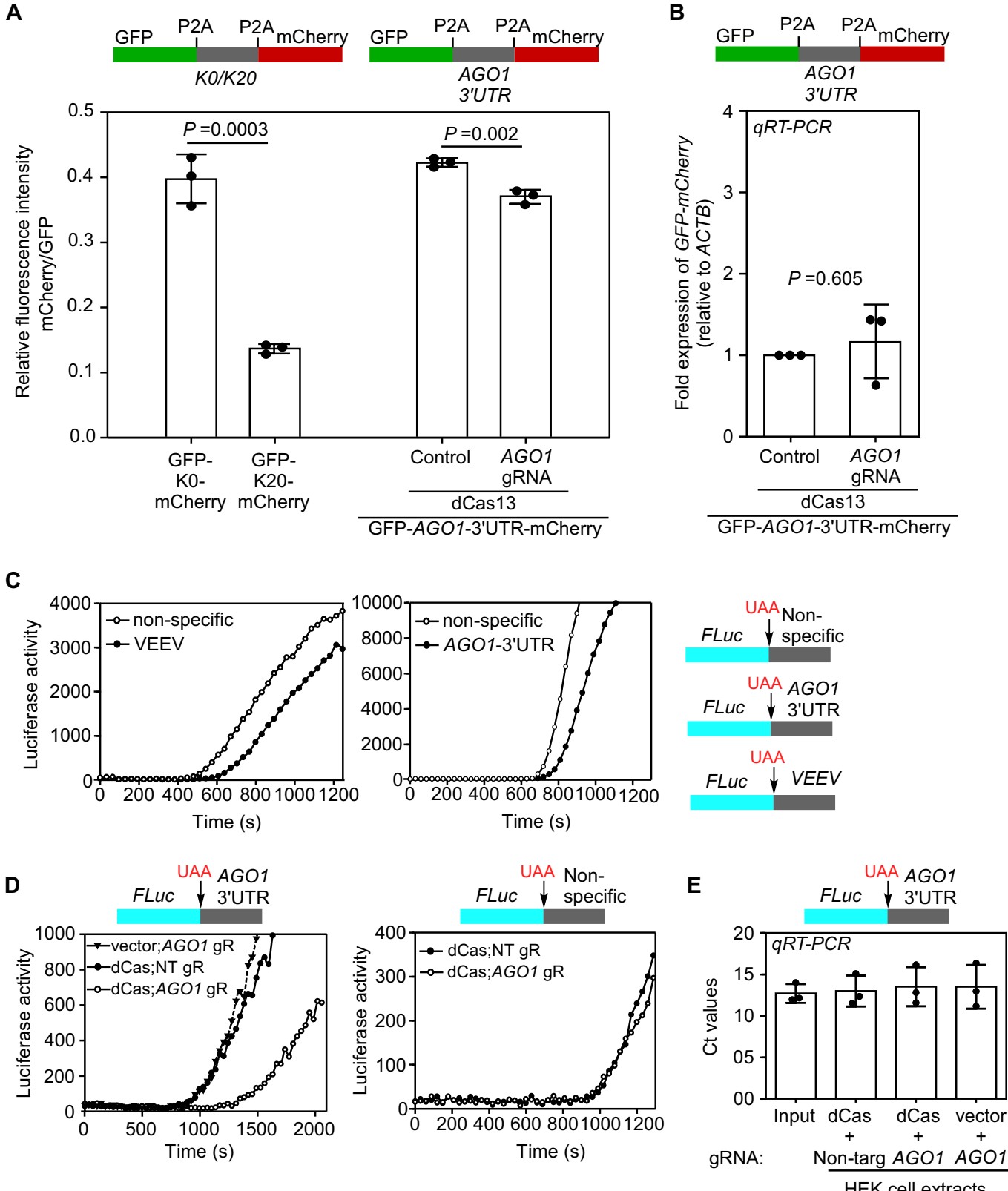

**Figure 4. AGO1-3′UTR-targeting dCas13 causes ribosomal pausing.**

(A) Fluorescence-based ribosomal pausing assay. HeLa cells were transfected with a plasmid construct containing cDNAs of GFP and mCherry separated by the proximal 3′UTR of *AGO1* (99 nucleotides). The graph shows the ratio of the mean fluorescence intensity of mCherry to that of GFP in the presence of dCas13 and *AGO1*-targeting gRNA. GFP-mCherry constructs with 20 lysine codons (AAA) between them were used as a positive control for the assay. (B) qRT-PCR analysis of the expression of GFP-*AGO1*-3′UTR-mCherry construct. (C) Luminescence-based ribosomal pausing assay. cDNA of FLuc was tagged with the proximal 3′UTR of *AGO1* (99 nucleotides) or a non-specific sequence of same length or the SCR-inducing sequence of Venezuelan equine encephalitis virus (VEEV). They were in vitro transcribed, and the resulting mRNA was in vitro translated using rabbit reticulocyte lysate. FLuc activity (luminescence) was measured every 30 s. (D) Luminescence-based ribosomal pausing assay performed in the presence of extracts from HEK293 cells expressing dCas13 and *AGO1*-targeting gRNA or nontargeting gRNA. All graphs are representatives of three independent experiments. (E) Ct values obtained by qRT-PCR showing the levels of in vitro transcribed FLuc mRNA (used in the assay shown in in (D)) incubated with HEK293 cell extracts for 75 min. Data information: Graphs in (A, B, E) show mean ± SD, $N = 3$ experiments. Two-sided Student's *t* test was used to calculate the *P* values. Source data are available online for this figure.

of the PTC) were transfected in HEK293 cells. While β-globin was not detected in cells expressing nontargeting gRNA, we could detect full-length β-globin-GFP protein in cells expressing dCas13 and *HBB*-targeting gRNA. *HBB*-targeting gRNA alone, without dCas13, was unable to show this effect (Fig. 6B). These results show that dCas13 can be programmed to induce SCR across the thalassemia-causing PTC in *HBB* mRNA.

Next, we tested this strategy in endogenous mRNAs with PTCs. We used Caco-2 cells (human colorectal adenocarcinoma cell line) which harbor a nonsense mutation in *TP53* gene (GAG to TAG or E204* mutation). These cells do not express p53 protein because of this mutation. We observed p53 expression in these cells when they were transfected with plasmids expressing dCas13 and a *TP53*-targeting gRNA (targets downstream of the PTC). The p53 expression was comparable to that induced by G418 (Geneticin), an aminoglycoside known to induce SCR (Fig. 6C). Furthermore, we observed a significant increase in the expression of *CDKN1A* (p21) and *BAX* mRNAs, which are known targets of p53-mediated transcriptional activation, in Caco-2 cells transfected with dCas13 and *TP53*-targeting gRNA (Zhang et al, 2017). These observations show that dCas13-gRNA complex can induce SCR and result in physiologically significant levels of p53 (Fig. EV5A).

Finally, we tested CRSIPR-dCas13 strategy in a cellular model of hereditary spherocytosis. This is a genetic disorder that affects erythrocyte membrane causing erythrocytes to become fragile due to spherical shape instead of a normal biconcave shape. It is the most common RBC membrane disorder (Kalfa, 2021). We developed a cellular model of hereditary spherocytosis in K562 cells (human erythroleukemia cell line). Using CRISPR–Cas9 method, we mutated *SPTA1* (encodes alpha-spectrin) gene at the nucleotide position 2671 to generate a PTC (CGA to TGA or R891* mutation) in K562 cells. This mutation was chosen as it is found in hereditary spherocytosis patients (Chonat et al, 2019; Rieneck et al, 2022). We confirmed the mutation in a clonal population by sequencing the genomic DNA of K562 cells. The mutant cells showed much reduced alpha-spectrin protein levels (~250 kDa) compared to the parental wild-type cells. It was not completely absent because K562 cells are triploid (https://www.atcc.org/products/ccl-243). Importantly, dCas13 and *SPTA1*-targeting gRNA (targets the region downstream of the PTC) induced higher expression of alpha-spectrin in these cells compared to cells transfected with nontargeting gRNA or another gRNA that targeted *SPTA1* away from the PTC (SPTA1[15]) (Fig. 6D). dCas13-induced recovery of the alpha-spectrin protein was about 80% of the alpha-spectrin levels in the parental wild-type cells. The efficiency of recovery is limited by the efficiency of delivery (i.e., transfectability of cells). It should be noted that even partial restoration of functional proteins (e.g.,

20–30% of normal dystrophin levels in DMD patients) can provide therapeutic benefits in many genetic diseases (Keeling et al, 2014).

## Discussion

Using multiple mRNAs—*AGO1*, *VEGFA*, *MTCH2*, *HBB*, *GFP*, *TP53*, and *SPTA1*—we have demonstrated that dCas13 can be programmed with the help of a gRNA to enhance (or induce) SCR across the canonical (or premature) stop codons. Importantly, this is achieved in a transcript-selective and stop codon-specific manner without altering the transcript level. This specificity provides a key advantage over the existing SCR-inducing strategies/molecules, which are largely non-selective. This approach can also be used to modulate natural SCR events and can serve as a tool to understand the functional significance of SCR events. This is important because other available tools such as siRNAs cannot be used to understand the function of SCR events; siRNAs bring down the mRNA levels resulting in downregulation of both the canonical isoform and the SCR product. Thus, the induction of SCR is a new addition to the CRISPR-Cas system's expanding arsenal of biotechnological applications.

Our results show the ability of dCas13-gRNA to induce SCR across PTCs. We have demonstrated this in *GFP*, *HBB*, *TP53*, and *SPTA1* mRNAs by western blotting, which detects SCR products of expected molecular weight. There was no significant increase in these mutant mRNA levels in the presence of dCas13 and their specific gRNAs, which rules out any role of mRNA levels in the observed SCR induction in these cells.

This property of dCas13 can be utilized in the treatment of diseases caused by nonsense mutations such as β-thalassemia, Duchenne muscular dystrophy, hereditary spherocytosis, etc. Since gRNAs ensure that dCas13 targets a specific mRNA, this strategy can achieve high specificity unlike other SCR-inducing agents such as ataluren and aminoglycoside. Also, this method can be explored for use in combination with other SCR-inducing agents to improve efficiency and reduce their toxicities. However, this method has to be personalized to a patient depending on the location of the nonsense mutation. This necessitates validation of the method for every PTC, which might increase the cost. Another challenge, as in case of all CRISPR-based therapeutic strategies, is the delivery of dCas13 and gRNAs in vivo. Viral vectors (adenovirus, adeno-associated virus, and lenti virus), lipid-based nanoparticles, and engineered extracellular vesicles, which are being explored as delivery methods for other CRISPR-based therapies, can be considered for the delivery of dCas13 and gRNAs (Madigan et al, 2023; Yip, 2020).

We acknowledge that expressing the wild-type version of the mRNA with a disease-causing PTC may be a better approach than

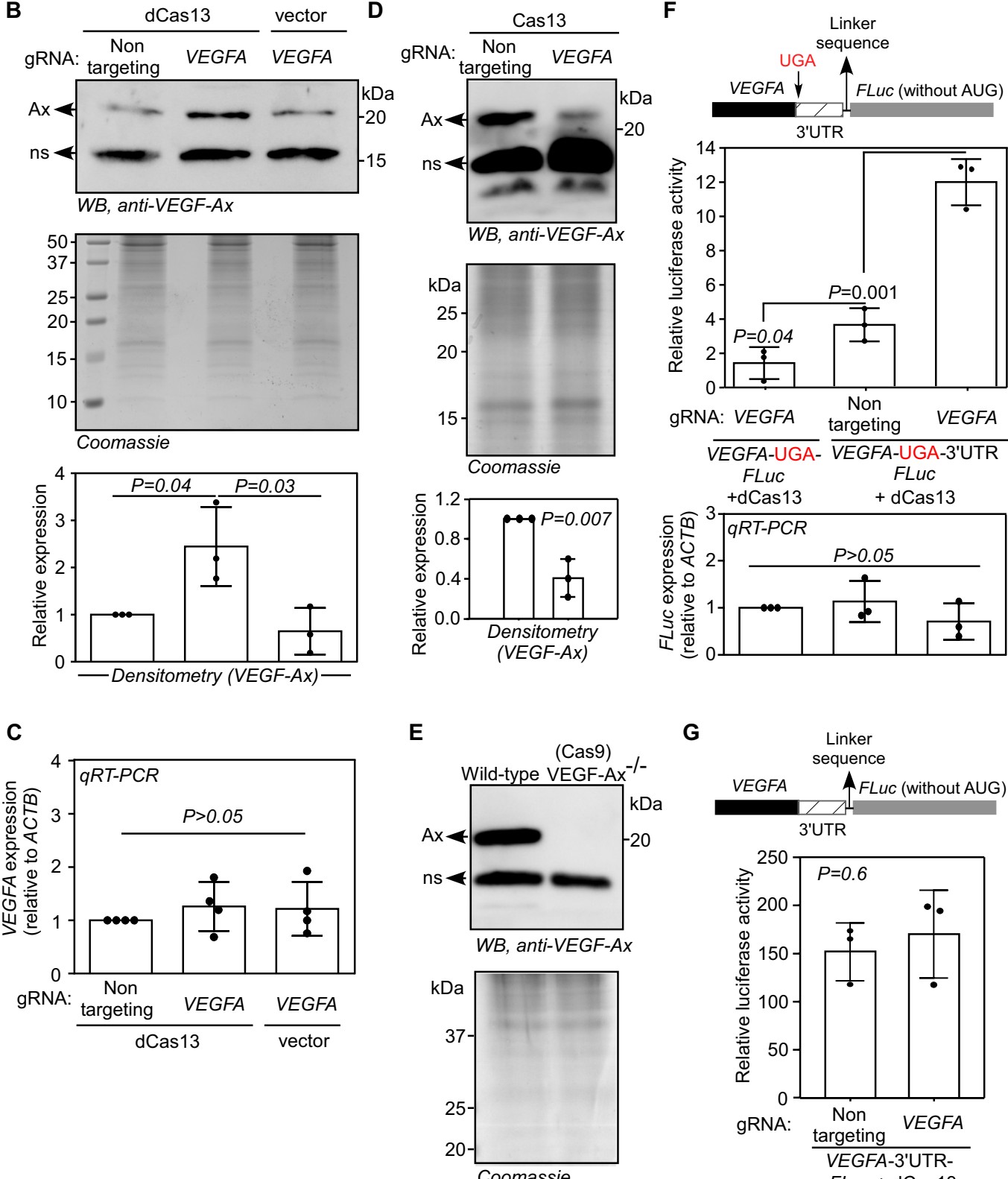

**Figure 5.  *VEGFA*-targeting dCas13 enhances SCR of *VEGFA* mRNA.**

(**A**) The sequence of the proximal 3'UTR of *VEGFA*. The region targeted by the gRNA is shown in blue. The canonical stop codon (UGA) is shown in red. (**B**) Western blot showing the effect of dCas13 on the SCR of *VEGFA*. HEK293 cells were transfected with plasmids expressing dCas13 and *VEGFA*-targeting gRNA or nontargeting gRNA. The conditioned medium was collected 48 h after transfection and subjected to western blotting to detect VEGF-Ax (the SCR product of *VEGFA*) expression. (**C**) qRT-PCR result showing the expression of *VEGFA* mRNA in those cells. (**D**) Western blot showing the level of VEGF-Ax in the conditioned medium of HEK293 cells expressing Cas13 and *VEGFA*-targeting gRNA. The conditioned medium was collected 48 h after transfection. (**E**) Western blot showing the absence of VEGF-Ax in the conditioned medium of VEGF-Ax knockout cells generated using CRISPR–Cas9 system. Ax, VEGF-Ax; ns, non-specific band. (**F**) Luminescence-based SCR assay. The cDNA of firefly luciferase (FLuc) was cloned downstream of and in-frame with the cDNA of *VEGFA* along with its proximal 3'UTR (63 nucleotides) such that luminescence is expected only after SCR (schematic). The constructs were transfected in HEK293 cells along with plasmids expressing dCas13 and *VEGFA*-targeting or nontargeting gRNA. The luminescence was measured 24 h after transfection. FLuc activity relative to the activity of the co-transfected *Renilla* luciferase is shown. The bottom graph shows the expression of *FLuc* mRNA. (**G**) Effect of dCas13 on canonical translation. *VEGFA*-3'UTR-FLuc construct without any stop codon in between was transfected in HEK293 cells along with plasmids expressing dCas13 and gRNAs. Relative luciferase activity was measured as described above. Data information: Graphs in (**B–D**, **F**, **G**) represent mean ± SD, $N = 3$ (or 4 in (**C**)) experiments. Two-sided Student's t test was used to calculate the *P* values. Source data are available online for this figure.

inducing SCR by expressing dCas13-gRNA complex for the treatment of certain genetic diseases. However, this is not feasible in several genetic diseases. For example, the dystrophin mRNA (*DMD*, NM_004006.3), which is mutated in Duchenne Muscular Dystrophy, has 11058 nucleotides-long coding sequence. It is a technical challenge to deliver and express such a large mRNA. In some conditions, PTCs result in truncated proteins with dominant-negative functions, which contribute to the pathology. For example, SOX10 mutations in Waardenburg-Shah syndrome (Sham et al, 2001). In such conditions expressing the healthy version of the protein might not provide therapeutic benefits. However, induction of SCR will reduce truncated proteins, and therefore pathologies associated with it. Expression of certain proteins (e.g., MeCP2) beyond physiological levels can lead to pathological conditions (Collins et al, 2004). For example, mutations in MeCP2 result in Rett's syndrome. In such cases, it is very difficult to express the healthy version of the *MECP2* mRNA without risking the overexpression and associated adverse effects. Furthermore, it is difficult to determine the optimal level of expression as the expression of a gene varies across cell types and tissue types. This problem will be minimized if we employ induction of SCR as they act on already expressed mRNAs according to the needs of the cells and the tissues. Thus, at least in the conditions described above, induction of SCR across PTCs seems to be a better strategy than expressing the mRNA itself. Our study demonstrates a transcript-selective strategy based on dCas13-gRNA to achieve this.

## Methods

### Reagents and tools table

| Reagent/resource | Reference or source | Identifier or catalog number |
|---|---|---|
| **Experimental models** | | |
| HEK293 cells (*H. sapiens*) | Manjunath et al, 2020 | |
| HeLa cells (*H. sapiens*) | Singh et al, 2019 | |
| Caco-2 cells (*H. sapiens*) | A kind gift from Prof. Utpal Tatu, Department of Biochemistry, IISc | |
| K562 cells (*H. sapiens*) | National Center for Cell Sciences, Pune, India | |

| Reagent/resource | Reference or source | Identifier or catalog number |
|---|---|---|
| **Recombinant DNA** | | |
| *AGO1*-UGA-*FLuc* | Singh et al, 2019 | |
| *AGO1*-UGA-3'UTR-*FLuc* | Singh et al, 2019 | |
| *AGO1*-3'UTR-*FLuc* | Singh et al, 2019 | |
| *AGO1*-UGA-*FLAG-HA* | This study | |
| *AGO1*-UGA-3'UTR-*FLAG-HA* | This study | |
| *AGO1*-3'UTR-*FLAG-HA* | This study | |
| *MTCH2*-UGA-*Fluc* | Manjunath et al, 2020 | |
| *MTCH2*-UGA-3'UTR-*FLuc* | Manjunath et al, 2020 | |
| *MTCH2*-3'UTR-*FLuc* | Manjunath et al, 2020 | |
| *VEGFA*-UGA-*FLuc* | Eswarappa et al, 2014 | |
| *VEGFA*-UGA-3'UTR-*FLuc* | Eswarappa et al, 2014 | |
| *VEGFA*-3'UTR-*FLuc* | Eswarappa et al, 2014 | |
| *HBB*[w16*]-GFP | Kar et al, 2020 | |
| FLAG-HA-GFP[w57*] | modified from pMHG-W57* reporter from Addgene #72850 | |
| GFP-K0-mCherry | Juszkiewicz and Hegde, 2017 | |
| GFP-K20-mCherry | Juszkiewicz and Hegde, 2017 | |
| GFP-*AGO1*-3'UTR-mCherry | This study | |
| FLuc-*AGO1*-3'UTR | This study | |
| FLuc-non-specific | This study | |
| FLAG-HA-AGO1x | Singh et al, 2019 | |
| Fluc-*VEEV* | This study, Ref : Lashkevich et al, 2020 | |
| pC014-LwCas13a-msGFP | Addgene | #91902 |
| pC015-dLwCas13a-NF | Addgene | #91905 |
| pC016-LwCas13a guide expression backbone (with U6 promoter) | Addgene | #91906 |
| pHAGE-IRES-puro-NLS-dPguCas13b-EGFP-NLS-3xFLAG | Addgene | #132400 |
| pLentiRNACRISPR_001-hU6-DR_BsmBI-EFS-PguCas13b-NLS-2A-Puro-WPRE | Addgene | #138143 |

| Reagent/resource | Reference or source | Identifier or catalog number |
|---|---|---|
| pLentiRNACRISPR_001-hU6-DR_BsmBI-EFS-dPguCas13b-NLS-2A-Puro-WPRE | This study | |
| pSpCas9(BB)-2A-Puro | Addgene | #48139 |
| pSpCas9(BB)-2A-GFP | Addgene | #48138 |
| pcDNA3.1B | Invitrogen | V800-20 |
| pRL-SV40 | Promega | E2231 |
| pmCherry-c1 | A kind gift from Prof. P.N. Rangarajan Lab, Department of Biochemistry, IISc | |
| **Sequence-based reagents** | | |
| Guide RNAs | This study | Methods |
| sgRNAs | This study | Methods |
| PCR primers | This study | Methods |
| **Antibodies** | | |
| Anti-Ago1x | Singh et al, 2019 | |
| Anti-VEGF-Ax | Eswarappa et al, 2014 | |
| Anti-Ago1 | Novus Biologicals | NB100-2817 |
| Anti-GFP | BioLegend | 902602 |
| Anti-puromycin | Developmental Studies Hybridoma Bank | PMY-2A4 |
| Anti-p53 | Thermo Fisher | AHO0152 |
| Anti-FLAG | SIGMA | F1804 |
| Anti-Spectrin alpha-1 | Invitrogen | ARC1650 |
| Anti-HA | SIGMA | 11867423001 |
| Anti-actin HRP | SIGMA | A3854 |
| Goat anti-rabbit HRP | Invitrogen | 32460 |
| Goat anti-mouse HRP | Invitrogen | 32430 |
| Goat anti-mouse HRP | Jackson Immunoresearch | 115-035-003 |
| Donkey anti-rat HRP | Jackson Immunoresearch | 712-035-153 |
| **Chemicals, enzymes, and other reagents** | | |
| DMEM high glucose | HiMedia | AL007A |
| Penstrep | Lonza | 17-602E |
| Fetal bovine serum (FBS) | Gibco | 10270-106 |
| Lipofectamine 2000 | Thermo Fisher Scientific | 11668-019 |
| Phusion High-Fidelity DNA polymerase | Thermo Fisher Scientific | F530L |
| PNK | Thermo Fisher Scientific | EK0031 |
| Dual-Luciferase Reporter Assay System | Promega | E1910 |
| Cell culture microplate, 6-well | Corning Costar | 3516 |
| Cell culture microplate, 24-well | Corning Costar | 353047 |
| Cell culture microplate, 96-well | Corning Costar | 3599 |
| Assay plate, 96-well, white, flat bottom | Corning Costar | 3912 |

| Reagent/resource | Reference or source | Identifier or catalog number |
|---|---|---|
| TaKaRa Taq | TaKaRa | R001A |
| TB Green Premix Ex TaqII | TaKaRa | RR820A |
| Rabbit reticulocyte lysate | Promega | L4960 |
| anti-FLAG M2 Affinity Gel | SIGMA | A2220 |
| RevertAid RT | Thermo Fisher Scientific | EP0442 |
| RNA isoPlus | TaKaRa | 9109 |
| Puromycin | SIGMA | P8833 |
| D-luciferin, sodium salt | Gold Bio | LUCNA-1G |
| Ponceau S | Amresco | 860 |
| Trichloroacetic acid (TCA) | SIGMA | T0699 |
| Coomassie Brilliant Blue | Amresco | 472 |
| PVDF membrane | Merck | IPVH00010 |
| Clarity Western ECL Substrate | Bio-Rad | 1705061 |
| Femto ECL reagent | Giri Diagnostics | GDICS9004 |
| Protease inhibitor cocktail | Promega | G6521 |
| Protein Assay Dye Reagent | Bio-Rad | 5000006 |
| T7 RNA polymerase | Thermo Fisher Scientific | EP0111 |
| RiboLock | Thermo Fisher Scientific | E00381 |
| Paraformaldehyde | SIGMA | 158127 |
| T4 DNA ligase | Thermo Fisher Scientific | EL0011 |
| BbsI | Thermo Fisher Scientific | ER1011 |
| BsmBI | Thermo Fisher Scientific | ER0451 |
| Click-iT® HPG Alexa Fluor® Protein Synthesis Assay Kit | Life Technologies | C10428 |
| Tsp45I | Thermo Fisher Scientific | ER1511 |
| PrimeSTAR GXL DNA polymerase | TaKaRa | R050A |
| Polyethylenimine linear | Polysciences inc | 23966 |
| **Other** | | |
| Cytoflex LX | Beckman Coulter | |
| CFX96 real-time PCR system | Bio-Rad | |
| Glomax Explorer | Promega | |
| ChemiDoc Imaging System | Bio-Rad | |
| FACSAria II sorter | BD Biosciences | |
| 4D-Nucleofector X unit | Lonza | |

## Cell culture

HEK293, HeLa, and Caco-2 cells were cultured using Dulbecco's Modified Eagle's Medium (DMEM, HiMedia), which was supplemented with 10% fetal bovine serum (FBS, Gibco) and 1% antibiotics (10000 U/ml penicillin, 10000 μg/ml streptomycin, Lonza). The cells were incubated in a humidified atmosphere at 37 °C with 5% $CO_2$. K562 cells were grown in RPMI-1640 medium under the same

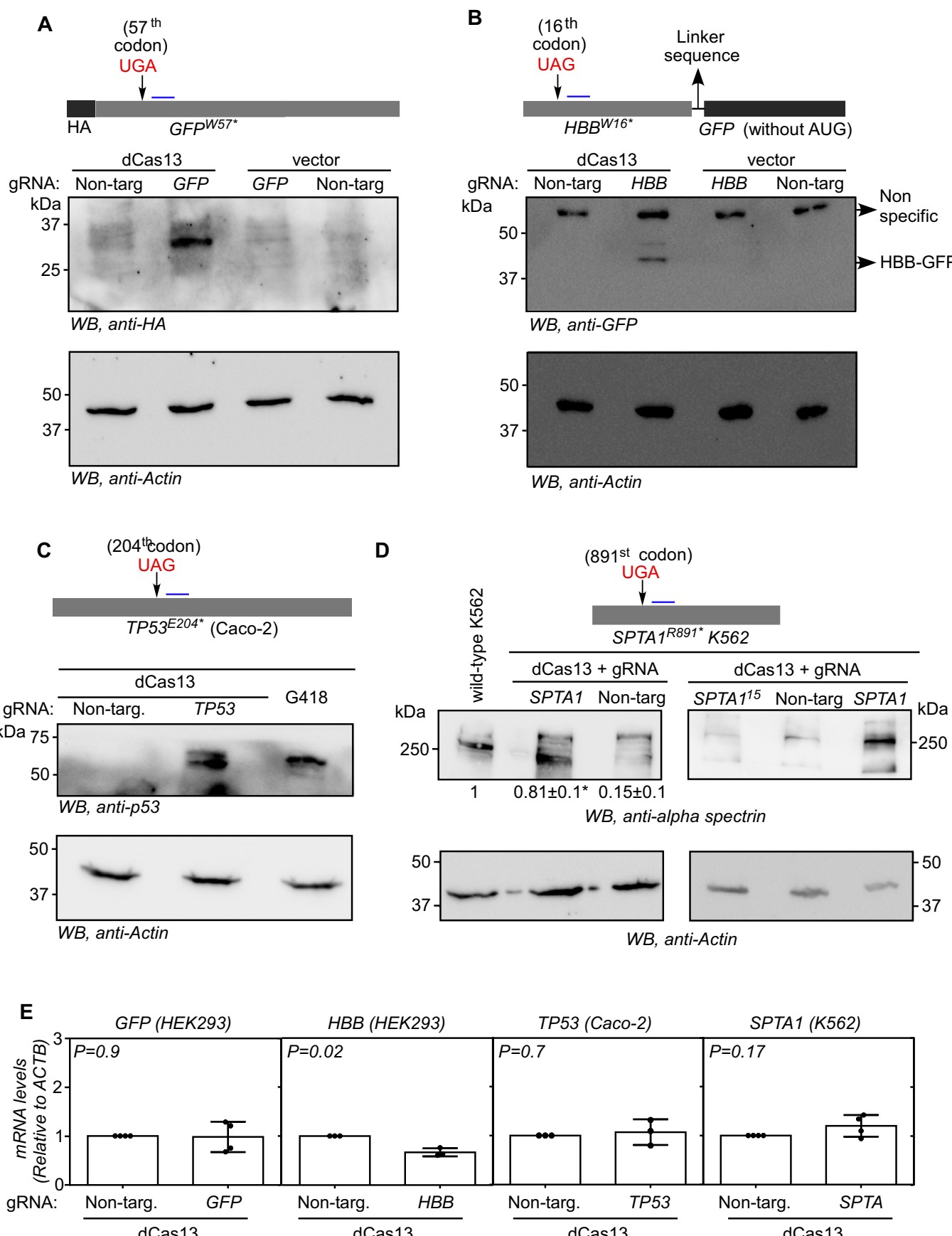

**Figure 6. Induction of SCR across premature termination codons (PTC) using dCas13.**

(A) Western blot showing the expression of full-length HA-tagged GFP from a construct with PTC at the 57[th] codon of GFP coding sequence. The construct was transfected in HEK293 cells along with plasmids expressing dCas13 and a GFP-specific gRNA or a nontargeting gRNA. (B) Western blot showing the expression of full-length GFP-tagged human β-globin from a construct with PTC at the 16th codon of *HBB*. The construct was transfected in HEK293 cells along with plasmids expressing dCas13 and a *HBB*-specific gRNA or a nontargeting gRNA. (C) Western blot showing the expression of full-length human p53 in Caco-2 cells whose genome has a PTC at 204th codon of *TP53*. Cells were transfected with plasmids expressing dCas13 and a *TP53*-specific gRNA or a nontargeting gRNA. G418, a known SCR-inducing agent was used (2 mg/ml) as a positive control. (D) Western blot showing the expression of full-length human alpha-spectrin in K562 cells whose genome was edited to create a PTC at 891st codon of *SPTA1* gene. Cells were transfected with plasmids expressing dCas13 and a *SPTA1*-specific gRNA or a nontargeting gRNA. *SPTA1*[15], gRNA targeting SPTA1 fifteen nucleotides downstream of PTC. Schematics indicate the approximate position of the PTCs and the gRNA target region (horizontal blue lines next to PTCs). Numbers below indicate mean ± SD densitometry values from three experiments. *$P = 0.01$ (*SPTA1* vs. nontargeting gRNA, two-sided Student's *t* test). All results are representatives of three independent experiments. (E) Results of qRT-PCR showing the expression of indicated mRNAs in cells transfected with plasmids expressing dCas13 and specific gRNAs or nontargeting gRNA. Data information: Bars, mean ± SD, $N = 3$ (*HBB* and *TP53*) $N = 4$ (*GFP* and *SPTA1*) experiments. Two-sided Student's *t* test was used to calculate the *P* values. Source data are available online for this figure.

conditions as above. Cell lines were authenticated by STR profiling. They were tested for mycoplasma contamination twice a year.

## Construction of plasmids

The reporter (Luciferase/GFP) constructs (in pcDNA3.1B vector background) used for SCR assays for *AGO1*, *VEGFA*, *MTCH2* and *HBB* have been described previously (Eswarappa et al, 2014; Kar et al, 2020; Manjunath et al, 2020; Singh et al, 2019). The FLAG-HA reporter constructs used for *AGO1* SCR assays were cloned in KpnI-BamHI in pcDNA3.1B vector backbone containing C-terminal FLAG-HA tag cloned in BamHI-NotI sites. In case of the GFP reporter for SCR across a premature termination codon, the GFP sequence with a nonsense mutation at the 57[th] codon was amplified from pMHG-W57* reporter, a gift from Daniel Liebler (Addgene plasmid #72850) (Halvey et al, 2012). It was cloned in pIRESneo-FLAG/HA plasmid to express the reporter with an N-terminal FLAG-HA tag. For the overexpression of full-length Ago1x with N-terminal FLAG and HA tags, the previously described FLAG-HA-AGO1x construct was used (Singh et al, 2019).

The GFP-K0-mCherry and GFP-K20-mCherry reporter constructs for ribosomal pausing/stalling assay were a kind gift from Prof. Ramanujan Hegde, MRC Laboratory of Molecular Biology, Cambridge (Juszkiewicz and Hegde, 2017). The proximal 3′UTR of *AGO1* (99 nucleotides) was cloned between the GFP and mCherry reporters. For the luciferase-based ribosomal pausing assay, the proximal 3′UTR of *AGO1* (99 nucleotides) was cloned downstream of the coding sequence of FLuc, separated by a stop codon in pcDNA3.1B vector backbone. As a length-matched control, a non-specific sequence was cloned in place of the *AGO1* proximal 3′UTR. As a positive control, the previously reported readthrough-inducing element from the Venezuelan Equine Encephalitis Virus (VEEV) was cloned downstream of the coding sequence of FLuc (Lashkevich et al, 2020).

pC014-LwCas13a-msfGFP plasmid expressing active Cas13, pC015-dLwCas13a-NF expressing the catalytically inactive mutant of Cas13 (dCas13) and pC016-LwCas13a guide expression backbone (with U6 promoter) were a gift from Feng Zhang (Addgene plasmids #91902, #91905 and #91906)(Abudayyeh et al, 2017). pHAGE-IRES-puro-NLS-dPguCas13b-EGFP-NLS-3xFlag vector, which expresses dCas13 from *Porphyromonas gulae*, was a gift from Ling-Ling Chen (Addgene plasmid # 132400)(Yang et al, 2019). pLentiRNACRISPR_001-hU6-DR_BsmBI-EFS-PguCas13b-NLS-2A-Puro-WPRE expressing PguCas13b was a gift from Neville Sanjana (Addgene plasmid # 138143) (Wessels et al, 2020). This was modified in this study to express dPguCas13 instead of PguCas13. Oligos were cloned in the guide expression backbones to

generate either the 28-nucleotide or 30-nucleotide gRNAs targeting the desired region.

The gRNA sequences are given below (5′ to 3′):

*AGO1*: GCTTTCTTCTATCCAGTGAGGTAACAGC/ AAGC TTTCTTCTATCCAGTGAGGTAACAGC

*MTCH2*: CTGGGACTACAGAAATGTCACTGTCCCT

*VEGFA*: CCCGAAACCCTGAGGGAGGCTCCTTCCT

*HBB* (W16*): GGGCCTCACCACCAACTTCATCCACGTT

*GFP* (W57*): CACCATAAGTCAGGGTGGTCACCAGAGT

*TP53* (E204*): CCACACTATGTCGAAAAGTGTTTCTGTC

*SPTA1* (R891*): CTGCTGGAACTGGACATTGGCTTCCAGA

Nontargeting (Abudayyeh et al, 2017): TAGATTGCTGTTC-TACCAAGTAATCCAT/ GTAGATTGCTGTTCTACCAAGTA ATCCATA.

## Luminescence-based SCR assays

HEK293 cells were seeded in 24-well plates at 70–80% confluency. In total, 200 ng/well of firefly luciferase-encoding reporter plasmids, 500 ng/well of dLwCas13a-NF and gRNA (gene-specific or nontargeting) were transfected using Lipofectamine 2000. Ten ng/well of *Renilla* luciferase was used as transfection control. Firefly and *Renilla* luciferase activities were measured using Dual-Luciferase Reporter Assay System (Promega) using GloMax Explorer (Promega) 24 h post transfection in case of samples involving *VEGFA* and *MTCH2*, and 48 h post transfection in case of *AGO1*.

For in vitro induction of SCR, the *AGO1*-Luciferase reporter construct was linearized using NotI restriction enzyme. Following this, the purified linearized plasmid was subjected to in vitro transcription using T7 RNA polymerase for 3 h at 37 °C. The RNA was purified, and the quality and concentration were determined using BioPhotometer (Eppendorf). The luciferase reporter RNA (500 ng) was in vitro translated using Rabbit Reticulocyte Lysate (Promega) in the presence of 4 μg cell extract (prepared as described under "Luminescence-based ribosomal pausing assay") and RNase inhibitor for 90 min. Luminescence was measured using the Luciferase Assay System (Promega) in the GloMax Explorer (Promega).

## Antibodies

Antibodies specific to the readthrough region of *VEGFA* (against the peptide AGLEEGASLRVSGTR) and *AGO1* (against the peptide RQNAVTSLDRRKLSKP) were generated as described in previous studies (Eswarappa et al, 2014; Singh et al, 2019). Anti-Ago1 antibody (Novus Biologicals, NB100-2817), Anti-GFP antibody (BioLegend,

902602), anti-puromycin antibody (PMY-2A4, Developmental Studies Hybridoma Bank), anti-p53 antibody (AHO0152, Thermo Fisher), anti-FLAG antibody (Sigma, F1804), anti-Spectrin alpha-1 antibody (Invitrogen, ARC1650), anti-HA antibody (Sigma, 11867423001), anti-Actin antibody (Sigma, A3854) and horseradish peroxidase-conjugated secondary antibodies (Thermo Fisher/Jackson Immunoresearch labs) were used as per the manufacturer's instructions.

## Western blotting-based SCR assays

HEK293 cells were seeded in 6-well plates at 70–80% confluency. For assays involving Cas13a-mediated knockdown of endogenous proteins, 2 µg/well of LwCas13a-NF along with 2 µg/well of gRNA-expressing plasmid (gene-specific or nontargeting) were transfected using Lipofectamine 2000. Twenty-four hours post transfection, the cell pellets (for Ago1x) or 48 h post transfection the conditioned medium (for VEGF-Ax) was harvested for western blotting. For dCas13-mediated induction of endogenous SCR products, either 3 µg/well of dLwCas13a-NF or 3 µg/well of dPguCas13b-3xFLAG, or 3 µg/well of pcDNA3.1B as vector control along with 2 µg/well of gRNA-expressing plasmid (gene-specific or nontargeting) were transfected using Lipofectamine 2000. Twenty-four hours post transfection, the cell pellets (for Ago1 and Ago1x) or 48 h post transfection the conditioned medium (for VEGF-Ax) was harvested for western blotting. The conditioned medium was subjected to trichloroacetic acid (TCA) precipitation before western blotting.

To carry out SCR assays with AGO1-FLAG-HA constructs, HEK293 cells were transfected with 2 µg/well of the reporter construct using Polyethylenimine (Polysciences). For experiments involving CRISPR-dCas13-mediated induction of SCR, HEK293 cells were transfected with 2 µg/well of *AGO1*-UGA-3′UTR-FLAG-HA along with 3 µg/well of dPguCas13b-gRNA construct (*AGO1*-targeting or nontargeting). After 48 h, the cells were lysed using RIPA lysis buffer and incubated with anti-FLAG M2 Affinity Gel (Sigma, A2220) resin beads overnight at 4 °C. The beads were then washed thrice and the immunoprecipitated proteins were extracted using Laemmli buffer. The samples were then subjected to western blotting using anti-HA antibody.

For the assays involving exogenous plasmids with PTCs, 3 µg/well of FLAG-HA-GFPw57*, 3 µg/well of dLwCas13a-NF or vector control along with 2 µg/well of gRNA-expressing plasmid (gene-specific or nontargeting) were transfected in HEK293 cells using Lipofectamine 2000 or Polyethylenimine. For the assay involving the premature stop codon in *HBB* mRNA, 500 ng/well of the GFP-encoding construct (*HBB^w16*-GFP*), 1 µg/well each of dLwCas13a-NF or vector control and gRNA-expressing plasmid (gene-specific or nontargeting) were transfected in HEK293 cells. The cell pellets were harvested 48 h post transfection.

For assays involving SCR across endogenous premature stop codon in *TP53*, Caco-2 cells were seeded in six-well plates at 80–90% confluency. They were transfected with 4 µg/well of dLwCas13a-NF along with 4 µg/well of gRNA-expressing plasmid (gene-specific or nontargeting) using Lipofectamine 2000. After 48 h, the cells were harvested. For assays involving SCR across the mutant premature stop codon in *SPTA1*, K562 cells were seeded in six-well plates at 70–80% confluence. They were transfected with 3 µg/well of dLwCas13a-NF along with 2 µg/well of gRNA-expressing plasmid (*SPTA1*-targeting or nontargeting) using Lipofectamine 2000. The cells were harvested 48 h post transfection.

Cell pellets were lysed in cell lysis buffer (20 mM Tris-HCl, 150 mM NaCl, 1 mM EDTA, 1% Triton-X with protease inhibitor cocktail (Promega)) and subjected to western blotting. Protein Assay Dye Reagent (Bio-Rad) was used to determine the protein concentration. The cell lysate (50–100 µg) was subjected to denaturing SDS-PAGE in 8% or 10% or 12.5% or 15% Tris-glycine gel. After the transfer of proteins onto a PVDF membrane (Merck), blocking was carried out (5% skimmed milk in PBS). Following this, the membrane was probed with the specific primary antibody and then with the respective horseradish peroxidase-conjugated secondary antibody. Clarity ECL reagent (Bio-Rad) or Femto ECL reagent (Giri Diagnostics) was used for the development of the blots, and the images were recorded using LAS-4000 imager (Fujifilm) or ChemiDoc Imaging System (Bio-Rad). Band intensities were quantified using ImageJ.

## RNA isolation, cDNA synthesis, and qRT-PCR

RNA isolation was carried out using RNAiso Plus (TaKaRa). The RNA quality and concentration were measured using BioPhotometer (Eppendorf). An equal amount of RNA (500 ng–1 µg) was used to carry out cDNA synthesis with oligo(dT) primers or gene-specific primers and RevertAid Reverse Transcriptase (Thermo Fisher Scientific). Quantitative real-time PCR was carried out using TB Green Premix Ex TaqII (TaKaRa) in CFX96 real-time PCR system (Bio-Rad). Melt curves were generated after each reaction. The amplified product was visualized using agarose gel electrophoresis to confirm the authentic amplification. The PCR conditions followed were as follows: 95 °C for 5 min, 40 cycles of 95 °C for 30 s, 55 °C/57 °C for 30 s, and 72 °C for 30 s, followed by a single final extension step at 72 °C for 5 min. Relative expression of the target mRNA relative to that of *ACTB* was calculated using the $2^{-\Delta\Delta Ct}$ method. Sequences (5′ to 3′) of primers are as follows:

FLuc: CAACTGCATAAGGCTATGAAGAGA; ATTTGTATTCAGCCCATATCGTTT

GFP: AAGTTCATCTGCACCACCG; TCCTTGAAGAAGATGGTGCG

*AGO1*: GGGAGCCACATATCGGGGCA/AAAGCCGTGCAGGTTCACCA; CTACCCCACCTCCCTCCTCCTTG

*VEGFA*: CTTGCCTTGCTGCTCTACC; CACACAGGATGGCTTGAAG

GFP^W57*: CTGGCAGACCATTACCAGCA; CTGCTGCTGTCACAAACTCC

*TP53*: AGGCCTTGGAACTCAAGGAT; TGAGTCAGGCCCTTCTGTCT

*HBB^W16*: TAATACGACTCACTATAGGG; ACCAACTTCATCCACGTT

*ACTB* (β-Actin): AGAGCTACGAGCTGCCTGAC; AGCACTGTGTTGGCGTACAG

*AGO1*-FLAG-HA: GACTACAAGGACGAC; TAGCGTAATCGGGCAC

*SPTA1*: GTAGAGGAAGGACACTTTG; ATCAGCACCATAGTTAGTAT.

## Global protein synthesis

### RiboPuromycylation

HEK293 cells were seeded in 6-well plates at 70–80% confluency. For CRISPR-dCas13-mediated induction of SCR in *AGO1*, 3 µg/well of dLwCas13a-NF or control plasmid along with 2 µg/well of gRNA-expressing (*AGO1*-targeting or nontargeting) plasmids were

transfected using Lipofectamine 2000. For Cas13a-mediated knock-down of *AGO1*, 2 μg/well of LwCas13a-NF along with 2 μg/well of gRNA-expressing plasmid (*AGO1*-targeting or nontargeting) were transfected using Lipofectamine 2000. For overexpression of Ago1x, either 4 μg/well of pcDNA3.1B or *FLAG-HA*-AGO1x was transfected using polyethylenimine. After 24 h or 48 h, cells were treated with 91 μM puromycin (Sigma) for 10–15 min at 37 °C. Following puromycin treatment, cells were lysed and 50 μg of the cell lysate was subjected to western blotting using anti-puromycin antibody. Ponceau S images of the PVDF membranes or Coomassie images of gels run with the same samples were used for normalization in densitometry analysis.

### Click-iT HPG Alexa Fluor protein synthesis assay

HEK293 cells were seeded in a 24-well plate at 75–90% confluence. dLwCas13a-NF or plasmid vector (750 ng/well) along with 500 ng/well of gRNA-expressing (*AGO1*-targeting or nontargeting) plasmids were transfected using Lipofectamine 2000. After 24 h, the Click-iT HPG Alexa Fluor protein synthesis assay (Life Technologies, C10428) was carried out as per the manufacturer's instructions. Briefly, cells were treated with 50 μM Click-iT HPG reagent in Methionine-free media for 30 min at 37 °C. The cells were then fixed using 4% paraformaldehyde and permeabilized using 0.5% Triton X-100 in PBS. In all, 1× Click-iT HPG reaction cocktail was then added for 30 min in the dark. The cells were washed with the Click-iT reaction rinse buffer. The samples were analyzed by flow cytometry using Cytoflex LX (Beckman Coulter).

## Dual-fluorescence-based ribosomal pausing assay

HeLa cells were seeded in 24-well plates at 70–80% confluence. 200 ng/well of the fluorescence reporter constructs along with 1 μg/well of dPguCas13b-gRNA construct were transfected using Lipofectamine 2000 following the manufacturer's protocol. After 24 h, the samples were subjected to flow cytometry analysis using Cytoflex LX (Beckman Coulter).

## Luminescence-based ribosomal pausing assay

HEK293 cells were seeded in six-well plates at 70–80% confluency. Two μg/well each of plasmids expressing dPguCas13 or plasmid vector and gRNA (either non-specific or gene-specific) were transfected using Lipofectamine 2000. After 48 h, the cells were resuspended in hypotonic extraction buffer (20 mM HEPES (pH 7.5), 10 mM Potassium acetate, 1 mM Magnesium chloride, and 4 mM DTT), homogenized at 3000 rpm using a motor-driven tissue grinder (Genetix), and kept on ice for 15 min. Samples were centrifuged at 15000 rpm, for 20 min at 4 °C and the supernatant was taken as the cell extract (described in (Zeenko et al, 2008)). The luciferase reporter constructs were linearized using XhoI and in vitro transcribed using T7 RNA polymerase (Thermo Scientific). Reporter RNA (100–250 ng) was in vitro translated using rabbit reticulocyte lysate in the presence of 4 μg of cell extract, 0.5 mM D-Luciferin, and RNase inhibitor. Luminescence was measured every 30 s using the GloMax Explorer (Promega).

## Assay to determine the efficiency of transfection

HEK293 cells were seeded in 24-well plates. The cells were transfected with combinations of pmCherry-C1 (250 ng/well), plasmids

expressing dPguCas13-GFP (500 ng/well) and *AGO1*-targeting gRNA (250 ng/well). After 48 h, the cells were harvested and subjected to flow cytometry using Cytoflex LX (Beckman Coulter).

## RNA immunoprecipitation

HEK293 cells were seeded in six-well plates at 70–80% confluency. Transfection was carried out with 3 μg/well of dPguCas13b-3xFLAG along with 2 μg/well of gRNA-expressing (nontargeting or gene-specific) plasmids. Forty-eight hours post transfection, the cells were washed with ice-cold PBS and fixed using 0.2% paraformaldehyde. After 15 min, the cells were treated with 125 mM glycine for 10 min to quench the crosslinking. The cells were washed with ice-cold PBS and lysed using RIPA lysis buffer with RNase inhibitor and Protease inhibitor cocktail. Cell lysates were incubated with anti-FLAG M2 Affinity Gel (Sigma, A2220) overnight with tumbling at 4 °C. Laemmli buffer was used to extract the immunoprecipitated proteins, which were subjected to western blotting. Immunoprecipitated RNA was extracted using TaKaRa RNAiso Plus. cDNA synthesis was carried out using oligo(dT) reverse primer. Quantitative real-time PCR was carried out using TB Green Premix Ex TaqII (TaKaRa) in CFX96 real-time PCR system (Bio-Rad). The PCR conditions: 95 °C for 5 min, 40 cycles of 95 °C for 30 s, 56 °C for 30 s, and 72 °C for 30 s, followed by a single final extension step at 72 °C for 5 min. The enrichment of *AGO1* mRNA in IP samples relative to the input samples was calculated using the $2^{-\Delta\Delta Ct}$ method. Sequences (5′ to 3′) of primers are as follows:

> *ACTB* (β-Actin): AGAGCTACGAGCTGCCTGAC; AGCACT GTGTTGGCGTACAG
> *AGO1*: GGGAGCCACATATCGGGGCA; CTACCCCACC TCCCTCCTCCTTG.

## CRISPR–Cas9-mediated deletion

sgRNAs (sequences given below) were cloned in pSpCas9 (BB)-2A-Puro plasmid. HEK293 or HeLa cells were transfected with these plasmids (two per gene) using Lipofectamine 2000 (Thermo Fisher Scientific). Twenty-four hours post transfection, transfected cells were selected using 2 μg/ml of puromycin (Sigma) for 5 days. The surviving cells were reseeded in a 96-well plate at a density of single cell per well. These clones were expanded and screened for the required genetic deletion by PCR using genomic DNA. The deletion was confirmed by sequencing of the PCR product and by western blotting.

> sgRNA sequences (5′ to 3′):
> *AGO1*: GCAGAACGCTGTTACCTCAC and GCTGTGCCACC CAAATCCAG
> *VEGFA*: ACAAGCCGAGGCGGTGAGCC and GGAAAGACT GATACAGAACG.

## CRISPR–Cas9-mediated mutation in *SPTA1* (R891*) in K562 cells

The point mutation (R891*) in the *SPTA1* gene was generated using CRISPR–Cas9 mediated gene editing wherein a single guide RNA (sgRNA) was used to generate double-strand breaks within the gene and this break was repaired by homology-mediated repair using a repair template containing the mutation of choice. Using

Benchling, an sgRNA (5′-GTCGCCTAGCAGCTCGAGCA-3′) that targeted the region close to the site of intended mutation was chosen and cloned in pSpCas9(BB)-2A-GFP (PX458) plasmid.

The repair template or homology donor was generated by PCR amplifying homologous regions, ~800 bp upstream and downstream of the site of editing, using genomic DNA from K562 cells and PrimeSTAR GXL DNA polymerase (TaKaRa). On either ends of the repair template, the sgRNA binding sequence along with its PAM motif was included (Forward primer: 5′-ATGCGGGCCCA-TAAGTCGCCTAGCAGCTCGAGCACGGTCACCCTCCAGGA TTTCCTT-3′; Reverse primer: 5′-ATGCGGATCCGTCGCCTAG-CAGCTCGAGCACGGAAAGCTGCTTTCCATCTATC-3′). PCR-based site-directed mutagenesis was carried out to convert the CGA (encoding arginine at 891st position of spectrin alpha protein) to TGA (stop codon). The PAM motif was also mutated from CGG to CTC, ensuring that the amino acid sequence is not altered. Codon-optimized silent mutations were made in the region corresponding to the sgRNA binding site and one codon downstream of the mutation site to prevent the repair template from being targeted by the sgRNA at the edit site (forward primer: 5′-CTCCGTGCTAGAGCCGCCAGGTGACAGAACGATCTGGA AGCCAAT-3′; reverse primer: 5′-ATTGGCTTCCAGATC GTTCTGTCACCTGGCGGCTCTAGCACGGAG-3′). The repair template was cloned in the reporter plasmid pmCherry-C1 between the enzyme sites ApaI and BamHI. Since the mCherry coding sequence in this plasmid did not have a stop codon of its own, this was included in the primers used for cloning to prevent the repair template from being translated along with mCherry.

Two micrograms of the sgRNA-Cas9-expressing plasmid and 5–6 μg of the repair template containing plasmid were co-transfected in K562 cells by nucleofection (4D-Nucleofector X unit, Lonza) as per the manufacturer's protocol. After nucleofection, the cells were seeded in a 35-mm dish. Forty-eight hours post transfection, cells expressing both GFP and mCherry (double positives) were sorted and seeded (one cell per well) in a 96-well plate using FACSAria II (BD) sorter. After expansion, genomic DNA was isolated from each clone and PCR amplification was done using primers flanking the site of mutation. The PCR product was then digested with Tsp45I (NmuCI) restriction enzyme whose site appears only after intended editing. PCR products of clones that had the CGA > TGA mutation were digested to obtain two products of 80 bp and 120 bp. The clones were further confirmed by Sanger sequencing (Appendix Fig. S5).

## Statistics

Two-sided Student's *t* test was used to test for the significance of the differences observed between samples in the experiments when samples showed normal distribution. Welch's correction was applied whenever the variance of the comparing sample groups was different. Information on sample size (3–4) used in the experiments is provided in the figure legends.

## Data availability

All the data related to this study are available in the manuscript.

## Peer review information

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

## Acknowledgements

The authors thank Dr. Paul L Fox, Cleveland Clinic Lerner Research Institute, USA, for sharing reagents. The authors thank the Flow Cytometry facility of the Indian Institute of Science. This work was supported by the Department of Biotechnology (DBT), India (grant no. BT/PR38405/GET/119/309/2020). SME is a recipient of the Swarnajayanti Fellowship (SB/SJF/2020-21/18) from the Department of Science and Technology (DST)-Science and Engineering Research Board (SERB), India. The authors gratefully acknowledge the financial support from STARS grant from the Ministry of Education (STARS/APR2019/BS/328/FS), DST Funds for Improvement of S&T infrastructure, and funds from the University Grants Commission, India provided to the Department of Biochemistry, Indian Institute of Science. LEM acknowledges the Research Associateship received from the Indian Council for Medical Research (ID-2021-13048).

## Author contributions

**Lekha E Manjunath**: Conceptualization; Data curation; Formal analysis; Investigation; Methodology; Writing—review and editing. **Anumeha Singh**: Investigation; Methodology. **Sangeetha Devi Kumar**: Resources; Methodology. **Kirtana Vasu**: Investigation; Methodology. **Debaleena Kar**: Resources. **Karthi Sellamuthu**: Resources; Methodology. **Sandeep M Eswarappa**: Conceptualization; Resources; Data curation; Formal analysis; Supervision; Funding acquisition; Investigation; Methodology; Writing—original draft; Project administration; Writing—review and editing.

## Disclosure and competing interests statement

SME and LEM are co-inventors in a patent related to the work described in this study (Indian Patent # 496848). The remaining authors declare no competing interests.

# Expanded View Figures

**Figure EV1.** (**A**) RT-PCR analysis showing the expression of Cas13 and dCas13 (*L. wadei*) in transfected HEK293 cells. Following primers were used (5′ to 3′): AGAACAACAAGGGCGAAGAGAAAT and CTTGCCTTCCAGTTCCAGGT. (**B**) dCas13 interacts with its target mRNA, *AGO1*, mediated by a specific gRNA. Constructs expressing dPguCas13b-3xFLAG along with gRNAs were transfected in HEK293 cells. Cell lysates were subjected to immunoprecipitation followed by RNA isolation and qRT-PCR to detect the enrichment of *AGO1* mRNA. Immunoprecipitates were also used for western blotting. Data information: Graph, mean ± SD, $N = 3$ experiments. (**C**) Schematics of constructs used in western blotting-based (FLAG-HA tag) and luminescence-based (firefly luciferase) SCR assays described in Fig. 1. (**D**) Western blotting-based SCR assay. The indicated plasmid constructs were transfected in HEK293 cells. After 48 h, they were subjected to immunoprecipitation followed by western blotting to detect FLAG-HA-tagged SCR product. (**E**) Luminescence-based SCR assay. The FLuc construct shown in (**C**) was transfected in HEK293 cells along with plasmids expressing dCas13 and *AGO1*-targeting or nontargeting gRNA. The luminescence was measured 48 h after transfection. FLuc activity relative to the activity of the co-transfected *Renilla* luciferase is shown. (**F**) Effect of CRISPR-dCas13 system on canonical translation and mRNA level. *AGO1*-3′UTR-*FLuc* construct without any stop codon in between was transfected in HEK293 cells along with plasmids expressing dCas13 and gRNA. Relative luciferase activity was measured as described above. qRT-PCR results in (**E, F**) show the expression of FLuc mRNA. Data information: Graphs in (**E, F**) are representatives of three independent experiments. Bars indicate mean ± SD ($N = 3$ Biological replicates). Two-sided Student's *t* test was used to calculate the *P* values. Source data are available online for this figure.

▶

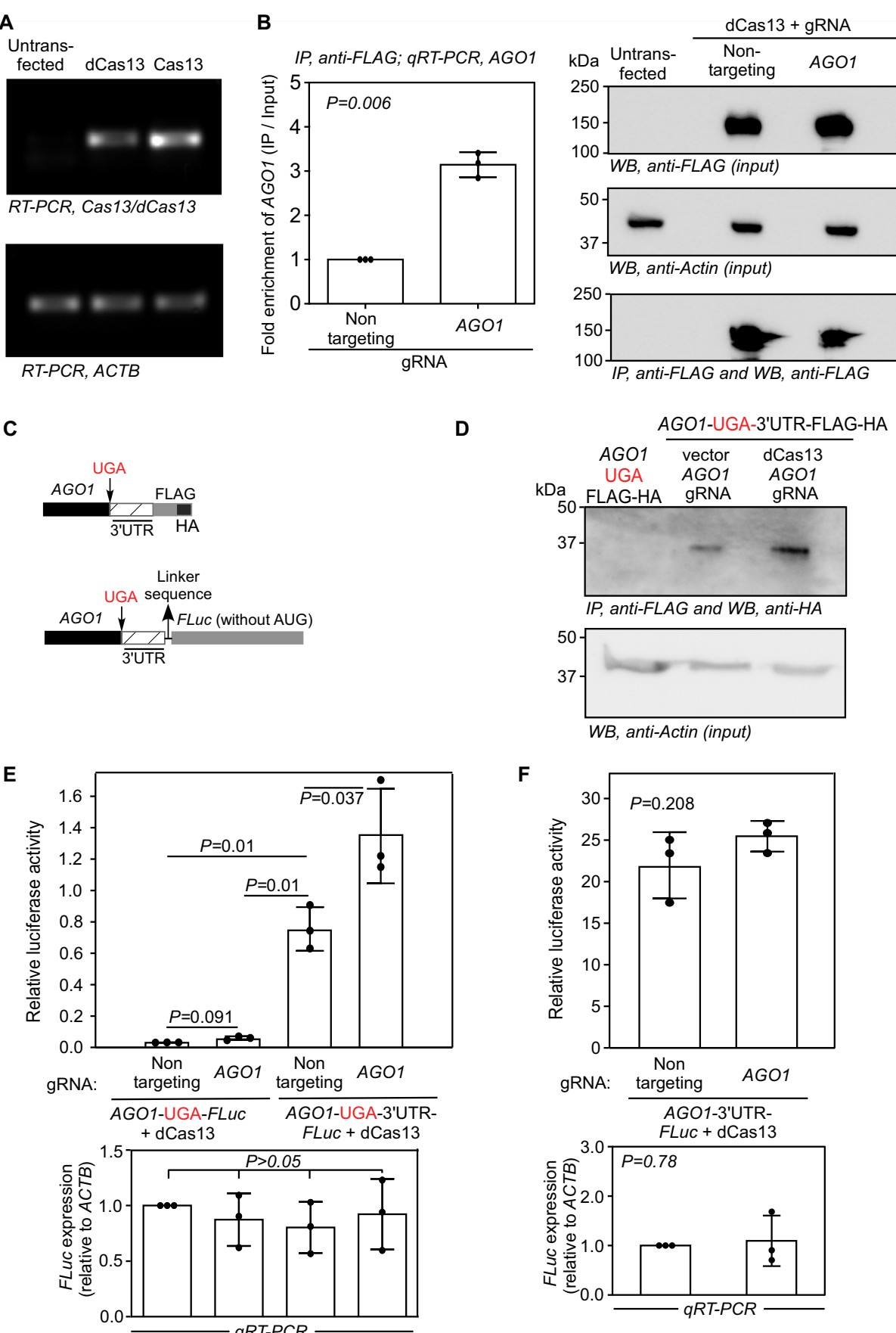

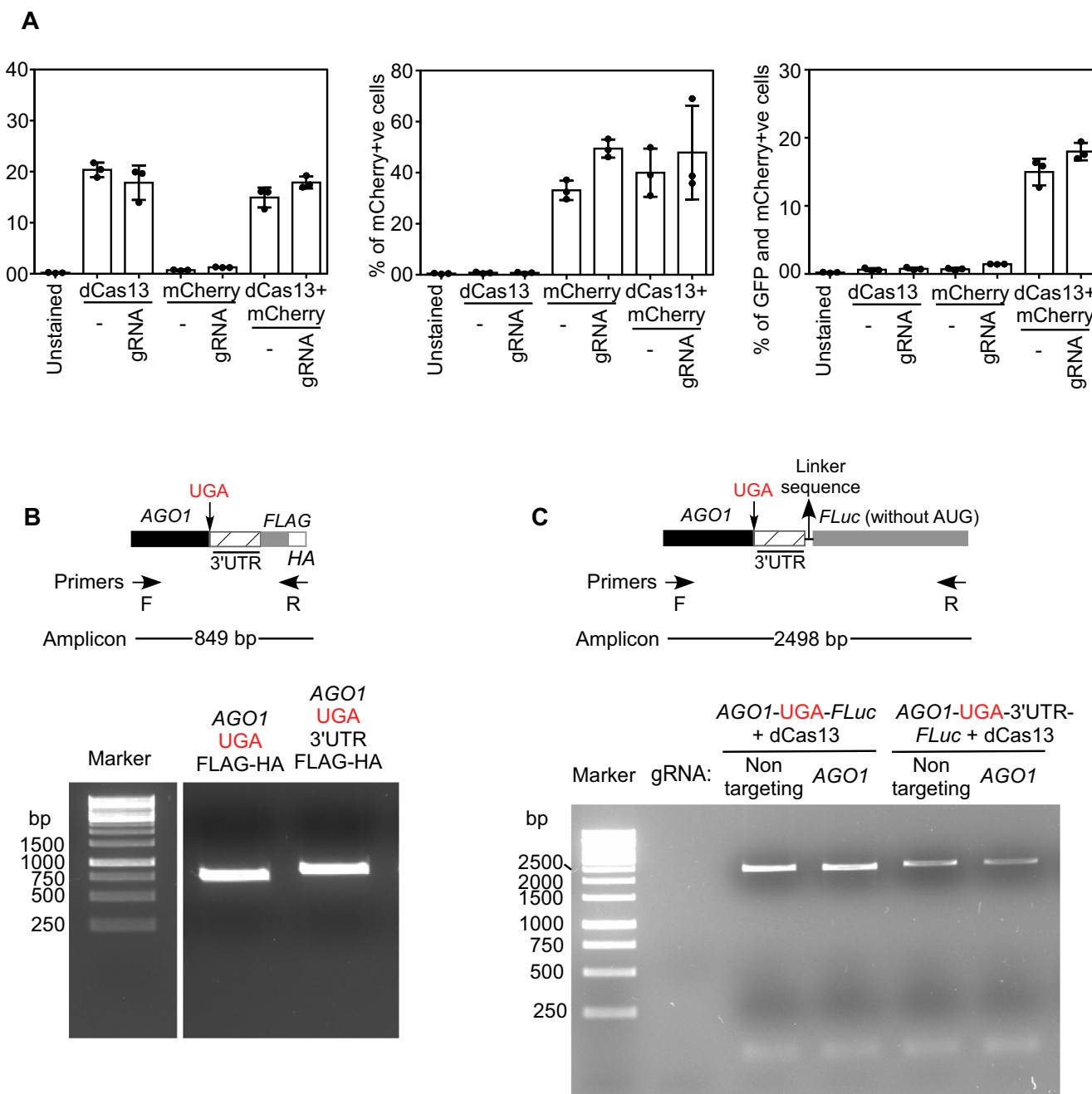

**Figure EV2.** (A) Evaluation of co-transfection efficiency. Constructs expressing dCas13 (also expresses green fluorescent protein), *AGO1*-targeting gRNA and mCherry (neutral reporter) were transfected in HEK293 cells. After 48 h, cells were subjected to flow cytometry analysis to quantify GFP+ve cells, mCherry+ve cells and double +ve cells. Graphs (mean ± SD, *N* = 3 biological replicates) are representatives of three independent experiments. (**B**, **C**) RT-PCR analysis of the transcripts expressed from the two reporter constructs used in SCR assays. Positions of the primers and the amplicon size are indicated. Following primers (5′ to 3′) were used for the RT-PCR: *AGO1*-HA: GTGCGGGTACAGCGACCACGGCAAGAG; TAGCGTAATCGGGCAC. AGO1-FLuc: GTGCGGGTACAGCGACCACGGCAAGAG; TTACAATTTGGACTTTCCG. Sequencing results of these transcripts are provided in Appendix Figs. S1 and S2. Source data are available online for this figure.

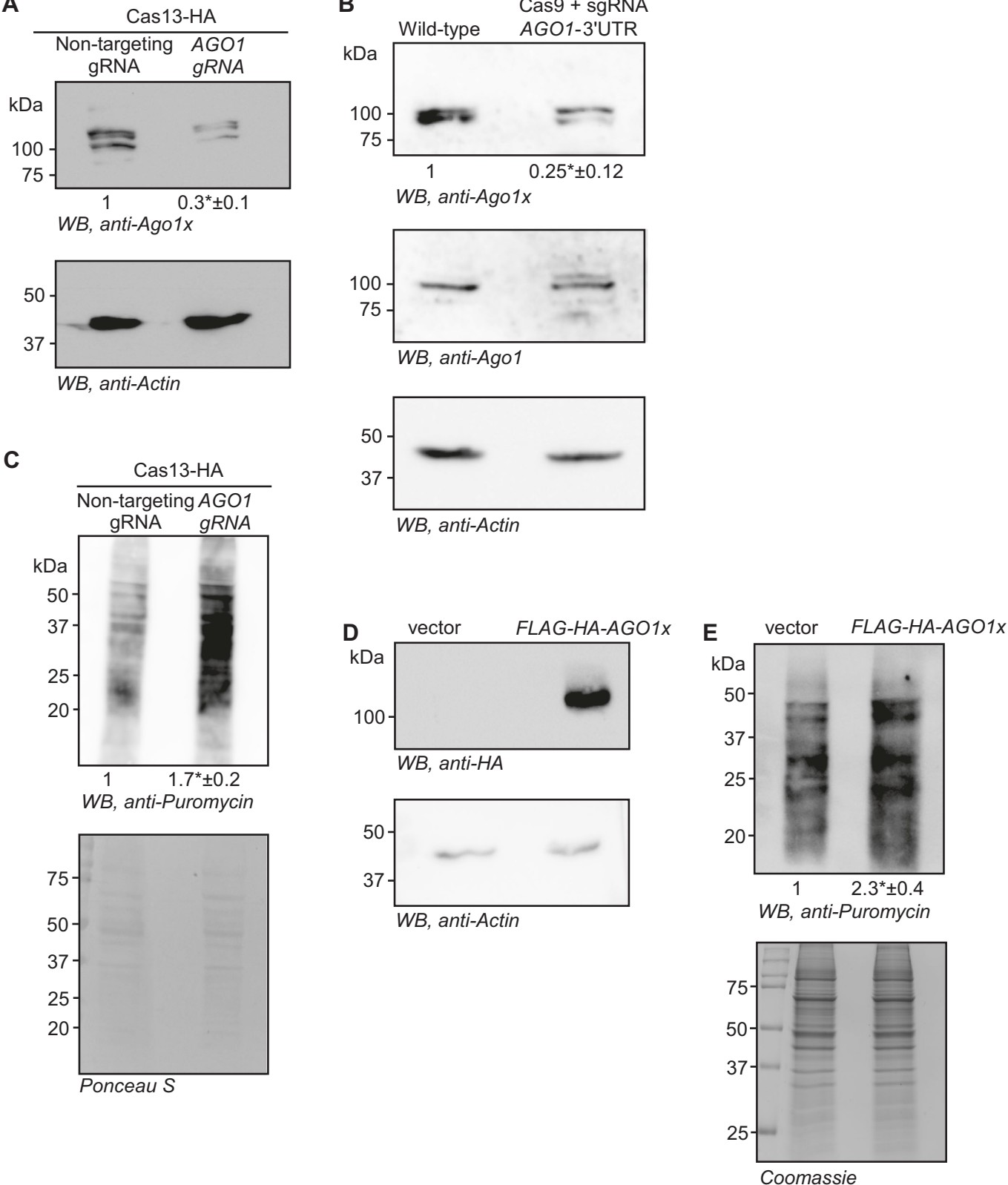

◀ **Figure EV3.** (**A**) Western blot showing reduced expression of Ago1x in HEK293 cells expressing Cas13 (catalytically active) and *AGO1*-targeting gRNA. Numbers indicate densitometry values (mean ± SD) from three experiments. *$P = 0.011$, two-sided Student's $t$ test. (**B**) Western blot showing expression of Ago1x and Ago1 in HeLa cells transfected with Cas9 and *AGO1*-3′UTR-targeting sgRNAs. Numbers indicate densitometry values (mean ± SD) from three experiments. *$P = 0.008$, two-sided Student's $t$ test. The sequencing of the genomic DNA PCR product revealed 51-nucleotide deletion in the proximal 3′UTR of *AGO1* (Appendix Fig. S3). (**C**) RiboPuromycylation assay performed in HEK293 cells expressing Cas13 (catalytically active) and *AGO1*-targeting gRNA. Numbers below indicate the densitometry values (mean ± SD, normalized to Ponceau staining) from three experiments. *$P = 0.04$, two-sided Student's $t$ test. (**D**) Western blot showing expression of exogenous FLAG-HA-tagged Ago1x in HEK293 cells. (**E**) RiboPuromycylation assay performed in HEK293 cells overexpressing FLAG-HA-tagged Ago1x in HEK293 cells. Numbers below indicate the densitometry values (mean ± SD, normalized to Coomassie staining) from three experiments. *$P = 0.028$, two-sided Student's $t$ test. Source data are available online for this figure.

**A**

ugaagauguggggagggacagugacauuucuguaguccccagaugcacagaa....

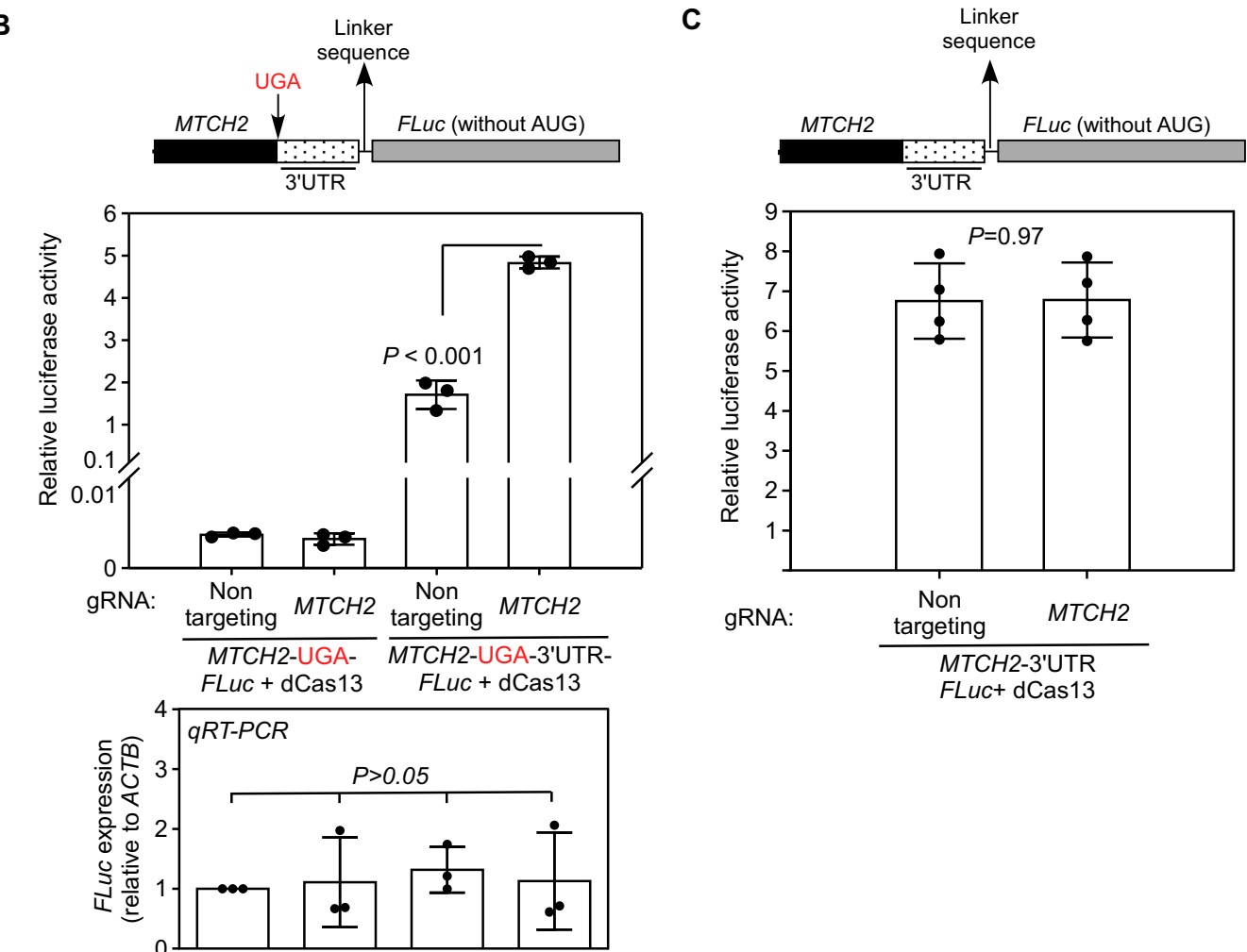

**Figure EV4. Enhancement of SCR across the canonical stop codon of *MTCH2* using CRISPR-dCas13 system.**

(A) The sequence of the proximal 3′UTR of *MTCH2*. The canonical stop codon (UGA) and the gRNA targeting region are in red and blue, respectively. (B) Luminescence-based SCR assay. The indicated constructs were transfected in HEK293 cells and firefly luciferase (FLuc) activity was measured 24 h after transfection. FLuc activity relative to the activity of the co-transfected *Renilla* luciferase is shown. Bottom panel shows the expression of *FLuc* mRNA measured by qRT-PCR. (C) Effect of CRISPR-dCas13 system on normal translation. *MTCH2*-3′UTR-FLuc construct without any stop codon in between was transfected in HEK293 cells along with plasmids expressing dCas13 and indicated gRNA. Relative luciferase activity was measured as described above. Data information: Graphs in (B, C) indicate mean ± SD, $N = 3$–4 experiments. Two-sided Student's *t* test was used to calculate the *P* value. Source data are available online for this figure.

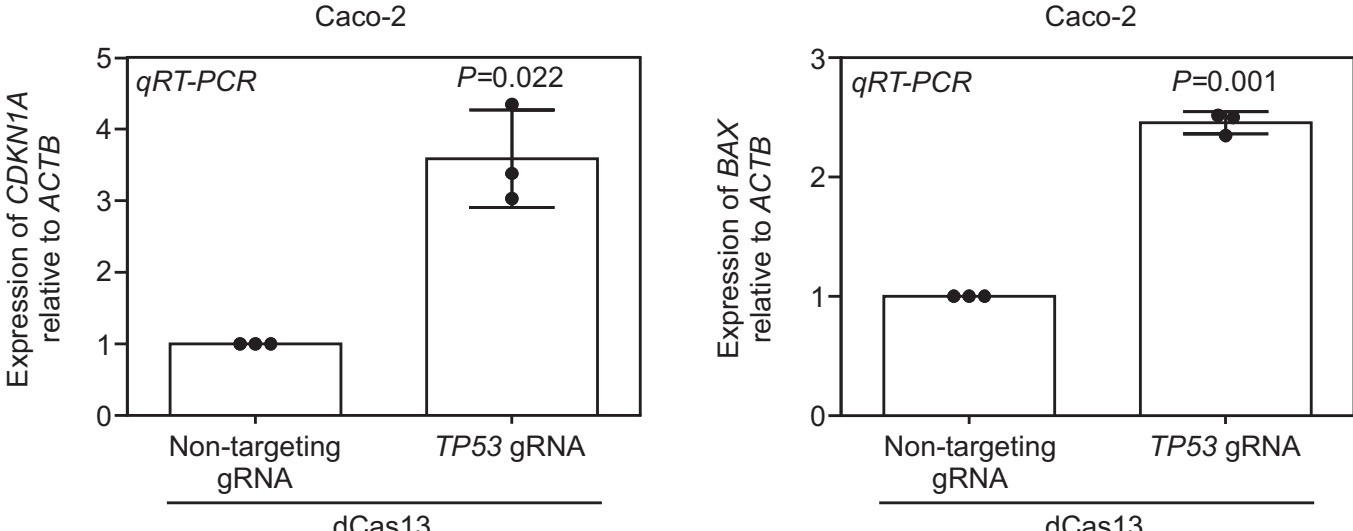

**Figure EV5.** qRT-PCR results showing the expression of *CDKN1A* (P21) and *BAX* in Caco-2 cells expressing dCas13 and the indicated gRNAs.

Graphs show mean ± SD ($N = 3$ technical replicates) and are representatives of three independent experiments. Two-sided Student's *t* test was used to calculate the *P* value. Following primers were used (5′ to 3′): *CDKN1A*: GGAAGACCATGTGGACCTGT and GGCGTTTGGAGTGGTAGAAA. *BAX*: GGGGACGAACTGGACAGTAA and CAGTTGAAGTTGCCGTCAGA. Source data are available online for this figure.

