## [Peer Review File · EMBO Reports]

Transcript-specific induction of stop codon readthrough using a CRISPR-dCas13 system

Lekha Manjunath, Anumeha Singh, Sangeetha Devi Kumar, Kirtana Vasu, Debaleena Kar, Karthi Sellamuthu, and Sandeep Eswarappa

Corresponding author(s): Sandeep Eswarappa (sandeep@iisc.ac.in)

Review Timeline:

Submission Date:	6th Mar 23
Editorial Decision:	17th Apr 23
Revision Received:	3rd Feb 24
Editorial Decision:	20th Feb 24
Revision Received:	23rd Feb 24
Accepted:	28th Feb 24

Editor: Esther Schnapp

Transaction Report:

Dear Prof. Eswarappa,

Thank you for the submission of your manuscript to EMBO reports. We have now received the full set of referee reports as well as referee cross-comments that are all pasted below.

As you will see, the referees acknowledge that the findings are potentially interesting. However, they also have several concerns, point out missing controls, that the advantage of this method as a therapeutic approach remains unclear, and that the effectiveness of the method for a disease-causing mutation in a relevant model system should be demonstrated. I think all referee concerns are reasonable and should be experimentally addressed, and this becomes clear also in the cross-comments. The only exceptions are points 1 and 2 from referee 1, which do not have to be addressed experimentally. If you wish, we can also discuss the revisions in a video chat.

I would thus like to invite you to revise your manuscript with the understanding that the referee concerns must be fully addressed and their suggestions taken on board. Please address all referee concerns in a complete point-by-point response. Acceptance of the manuscript will depend on a positive outcome of a second round of review. It is EMBO reports policy to allow a single round of major revision only and acceptance or rejection of the manuscript will therefore depend on the completeness of your responses included in the next, final version of the manuscript.

We realize that it is difficult to revise to a specific deadline. In the interest of protecting the conceptual advance provided by the work, we recommend a revision within 3 months (18th Jul 2023). Please discuss the revision progress ahead of this time with the editor if you require more time to complete the revisions.

- 1) A data availability section providing access to data deposited in public databases is missing. If you have not deposited any data, please add a sentence to the data availability section that explains that.
- 2) Your manuscript contains statistics and error bars based on $n=2$. Please use scatter blots in these cases. No statistics should be calculated if $n=2$.

3) We replaced Supplementary Information with Expanded View (EV) Figures and Tables that are collapsible/expandable online. A maximum of 5 EV Figures can be typeset. EV Figures should be cited as 'Figure EV1, Figure EV2' etc... in the text and their respective legends should be included in the main text after the legends of regular figures.

5) a complete author checklist, which you can download from our author guidelines <https://www.embopress.org/page/journal/14693178/authorguide>. Please insert information in the checklist that is also reflected in the manuscript. The completed author checklist will also be part of the RPF.

6) Please note that all corresponding authors are required to supply an ORCID ID for their name upon submission of a revised manuscript (<https://orcid.org/>). Please find instructions on how to link your ORCID ID to your account in our manuscript tracking system in our Author guidelines

<<https://www.embopress.org/page/journal/14693178/authorguide#authorshippinguidelines>>

I look forward to seeing a revised form of your manuscript when it is ready.

Yours sincerely,

Referee #1:

Manjunath et al described a CRISPR-based method to site-specifically promote premature termination codon (PTC) readthrough. They programmed dCas13 to bind to a mRNA region downstream to a PTC, thus presumably interfering with the translation termination process at the PTC by steric hindrance and in turn promoting readthrough. The approach may have therapeutic value for a broad range of genetic diseases caused by nonsense mutations. Data generally support the conclusions, and the manuscript is well written and presented. However, as a potential therapeutic approach, the reported technique can be strengthened by additional data.

1. Are dCas13 and a readthrough agent (e.g., G418 or suppressor tRNA) additive or synergistic for promoting readthrough? Although such a readthrough agent tends to cause off-target readthrough at normal stop codons, a combinatorial approach may further enhance PTC readthrough.
2. A dose-response experiment for dCas13 and gRNA to show which component may be limiting will be informative. This will inform on future optimization of the technique.
3. I suggest the authors discuss on the delivery methods for their technique in a therapeutic setting, particularly in the case of in vivo delivery.

Referee #2:

In this manuscript, the authors utilize the RNA targeting specificity of the CRISPR/Cas13 system and identify its novel application in elevating stop codon readthrough (SCR) in mammalian genes that naturally undergo SCR. What sets this system apart from other readthrough-inducing molecules, such as aminoglycosides, marcolides, atalurens, etc, is its ability to target select transcripts without disturbing global translation. By using specific guide RNAs, the authors direct a catalytically inactive dCas13 variant to regions downstream of the stop codon being investigated and perform a combination of readthrough assays to quantify SCR on exogenous reporter constructs as well as endogenous genes. They extend the efficacy of this system to induce SCR in premature termination codons in several other genes which can be crucial for developing therapeutics for genetic diseases caused by nonsense mutations. The authors attribute the SCR inducing potential of their system to transient ribosomal pausing caused by a molecular roadblock of gRNA-dCas13 complex.

While the authors present compelling data to support their claim, many of their results lack control experiments that are crucial to validate the role of CRISPR/Cas13 in inducing SCR. There are several points that need to be addressed by the authors before acceptance.

Specific comments:

- 1) Page 5, Paragraph 1, last line: The authors mention the gRNAs target mRNAs, "proximally downstream (4-20 nucleotides) of the stop codon". However, the schematics in Fig 1A, 5A, EV2 and the gRNA sequence list (Materials and methods, Page 16) indicate otherwise. Do the authors mean 4-20 codons?
- 2) Page 6, line 18: The authors claim that the gRNA and dCas13 work in a transcript-specific manner to enhance SCR. To verify that both molecules act in concert, the authors need to provide control experiments (Fig 1B, 1C) where the cells are transfected with the luciferase reporter construct AGO1-UGA-3'UTR-Fluc and the AGO1 3'UTR targeting gRNA, but not the dCas13 construct. The same comment also applies to the following experiments indicated in Fig 4B, 5B, 6A and 6B
- 3) As an addition to point 2, it is essential to estimate the co-transfection efficiency of the reporter and the dCas13 constructs to uncouple the effects from gRNA alone and gRNA+dCas13 that might render variability in the readout. Additionally, the co-transfection efficiency of renilla luciferase used to normalize Fluc activity might also attribute to the variability in the readout. Having some representative values for absolute readouts of firefly as well as renilla luciferase in a supplementary table would be helpful.
- 4) Fig 1B, 1C: Electrophoretic-based RT-PCR cannot be used as a quantitative method to assess differences in expression of transfected genes as the observed signal saturates after multiple PCR amplification cycles. If the RT-PCR is done under limiting conditions where the amplification of cDNA of interest is relatively linear, then it should be mentioned in the materials and methods section. RT-qPCR analysis would be a more sensitive method to negate any effects that might stem from differences in expression levels of transfected genes. Alternately quantitative western blots can also be used for this. This comment also applies to Fig 2B, 5C, EV1A, EV2B.
- 5) Page 7, Paragraph 1: How would the authors explain the fold differences in the enhancement of SCR by gRNA+dCas13 in MTCH2 (Fig EV2B) compared to AGO1 (Fig1B)
? It is also concerning that the relative luciferase activity of no-stop constructs for MTCH2 are so low in Fig EV2C.
- 6) Page 7, Paragraph 2: The western blot to detect Ago1x using anti-Ago1x, which is specific to the readthrough region of the protein, shows two bands that seem to increase upon induction by gRNA+dCas13 (Fig 2A,D) and three bands in Fig EV3. Similar experiments performed with dCas13(Pgu) seem to give only one band. Can the authors comment on this discrepancy?
- 7) pcDNA mentioned in Fig EV3 need to be explained in the text or legend.

8) Page 8, Paragraph 1: Utilization of CRISPR-Cas9 system to target the AGO1-3'UTR (Fig EV3B) may not only downregulate the expression of the SCR isoform Ago1x, but also the major protein Ago1 by compromising the transcript stability. The authors would need to elaborate on how the 3'UTR was modified with CRISPR/Cas9 and present a control blot with anti-Ago1 to assert their claim.

9) Page 9, Paragraph 1, 2nd last line: The authors report a 'significant' reduction in the ratio of the mean fluorescence intensity of mCherry to that of GFP, however, the graph (Fig 4A) shows only a modest reduction. The authors could change the verbiage.

Minor:

In general, there seems to be several inconsistencies with the brightness and contrast of western blot images throughout the manuscript, which the authors can work on minimizing. There also seems to be minor formatting differences with regards to the alignment of figure titles, positioning of axes labels (eg: Fig 1C RT-PCR, Fluc), capitalization etc, which the authors can work upon.

Referee #3:

The manuscript from Manjunath LE et al. aims at using dCAS13 to stimulate stop codon readthrough in an mRNA-specific way. The general idea is to target a downstream sequence of the stop codon to create a roadblock that would stall the ribosome at the stop codon. The idea is interesting although not completely original because several proteins have been already shown to play such role. The authors also explain this could be very interesting for stimulating PTC readthrough for therapeutics purposes. They are right to mention that currently there is no mRNA-specific way to stimulate readthrough, and that all current methods stimulate readthrough in a global way. However, I do not understand the rationale of the approach from a therapeutic point of view. Indeed, this approach would require introducing into the cells of the patients the dCAS13, and the gRNA. In this case, why not directly transform the cells with the WT mRNA corresponding to the defective gene, to restore a complete expression of the protein? This seems to me much more direct and efficient than trying to stimulate the readthrough and also probably subject to fewer harmful side effects.

All the work and reasoning are based on the use of these reporter systems, so it is essential that all the controls are correctly performed to eliminate alternative explanations. Actually, it is now well established that reporter systems suffer from many potential biases (alternative promoter, alternative splicing, leaky scanning, reinitiation...). They must be rigorously considered and excluded to support the working hypotheses; I am afraid this is not the case on this manuscript. It is curious to choose two genes for which the readthrough is programmed (Ago1 and VEGFA) to demonstrate that it is possible to stimulate it using dcas13. On Ago1 the authors have previously shown that there was a stimulation of readthrough efficiency by the small RNA let7-A which seems to bind to the region targeted by dcas13. The authors do not even discuss this possible competition and its consequences.

If the aim of the work is a therapeutic approach, why not have focused the approach on PTCs present in genetic diseases as at the end of the manuscript? for which it is unfortunately not clear if the induction efficiency will have a real impact or not.

I have several important comments for the authors:

-P6, L10 (figure 1B). a western-blot is necessary to confirm readthrough by visualising a fusion protein at the correct size. Currently the only conclusion is that the spacer region promotes Fluc activity. Many alternative mechanisms could explain the skipping of the stop codon (i.e alternative splicing removing the UAA codon, re-initiation downstream the UAA) with a so long insertion (>600nt) these events are likely to occur. A nanopore sequencing will be reassuring about the integrity of the mRNA and the absence of alternative promoters/initiators.

- Fig 1A is misleading because the grey region of ago1 is more than 600nt long, F-Luc is around 1kb, so please be proportional in the schematic.

Sequences would also be necessary in supplementary data to clearly understand which parts are present and removed between panels B and C figure 1. because this is certainly not only the blue part that is missing in panel C

- page 7, Figure 2A, 2B. Why anti-ago1x reveals two bands in panel A? this is not mentioned at all neither in the text nor in the legend of the figure. Why the readthrough form is not detected by anti-ago1 ?

at this stage, all changes in luciferase activities or protein level could be explained by alternative explanations. There is no direct evidence that dCas13 directly stimulates readthrough.

-P8, L3, there is no demonstration that the effect on translation observed comes from the higher synthesis of Ago1x (the readthrough form of ago1). To demonstrate the link they should use a siRNA against Ago1 to prove the direct link. Moreover there are more quantitative ways to quantify readthrough (polysome fractions or click-it kits coupled with alexa fluorophores) rather than ponceau staining and total puromycin staining.

P9, figure 4. the effects observed are very marginal, if any and can hardly explain the previously described translation defects (0.41 vs 0.37 probably)

Did the authors quantify mRNA level of the reporter? because ribosome stalling is known to induce mRNA degradation through the RQC pathway.

Moreover P2A sequences themselves induce a strong ribosome pausing (and drop off) (PMID : 25621616 + 8112307 + 28526819) this is far from a good idea to include them in a reporter system dedicated to quantify ribosomal pausing.

P10- Readthrough is probably less than 50% efficient in presence of dCas13 (although this is never properly quantified in the manuscript), especially by mixing RRL and HeLa cells extracts. So variations of Fluc activity due to RT is marginal in comparison to the luc activity generated from ribosomes that stopped at the UAA codon. Does mRNA level has been checked in RRL after the experiment to be sure the presence of gRNA of dCAS13 does not induce luc mRNA degradation?

P10, Fig 4B. Before concluding that they have demonstrated ribosome pauses, they should demonstrate that their system is capable of detecting pauses when bona-fide readthrough inducers are used. Controls are missing in this experiment.

Moreover, how can we deal with the previous model published by the authors indicating that Ago1x self-regulate its own readthrough using let7-a miRNA?

Why this is not shown? is there any competition between let7-a complex and dCas13?

P10- fig 5. Proper loading controls must be realized, Ponceau is not enough proportional to be relevant for quantification

P11 Fig5F. why not using gold standard dual reporter to quantify properly readthrough?

P12. Fig6. Some controls are also missing here. For example, there is no mRNA quantification. What about the possibility that dCas13 stabilize the mRNA? then background level of readthrough will be sufficient to see the protein by WB

P13 - Fig 6 do both panels of fig6C come from the same gel? G418 is used at which concentration?

Cross-comments from referee 1:

I agree with referee #3 that the advantages of site-specific readthrough as a therapeutic approach should be clarified, which would set it apart from other approaches such as gene replacement therapy. I also agree with referee #3 that demonstrating the effectiveness of the described method for a disease-causing mutation in relevant model systems would be more convincing.

Cross-comments from referee 3:

> Referee #1

>

> Manjunath et al described a CRISPR-based method to site-specifically promote premature termination codon (PTC) readthrough. They programmed dCas13 to bind to a mRNA region downstream to a PTC, thus presumably interfering with the translation termination process at the PTC by steric hindrance and in turn promoting readthrough. The approach may have therapeutic value for a broad range of genetic diseases caused by nonsense mutations. Data generally support the conclusions, and the manuscript is well written and presented. However, as a potential therapeutic approach, the reported technique can be strengthened by additional data.

>

> 1. Are dCas13 and a readthrough agent (e.g., G418 or suppressor tRNA) additive or synergistic for promoting readthrough? Although such a readthrough agent tends to cause off-target readthrough at normal stop codons, a combinatorial approach may further enhance PTC readthrough.

> 2. A dose-response experiment for dCas13 and gRNA to show which component may be limiting will be informative. This will inform on future optimization of the technique.

This point seems difficult to address, in my opinion. It is always difficult to evaluate the amount of dCas13 or gRNA that actually enters the cell.

> 3. I suggest the authors discuss on the delivery methods for their technique in a therapeutic setting, particularly in the case of in vivo delivery.

That could be interesting, however, as I wrote in my review, I do not believe in this approach as a real therapeutic solution, and I think this part makes the work look sexy and applied, but it's just for show.

>

>

> Referee #2

>

> In this manuscript, the authors utilize the RNA targeting specificity of the CRISPR/Cas13 system and identify its novel application in elevating stop codon readthrough (SCR) in mammalian genes that naturally undergo SCR. What sets this system apart from other readthrough-inducing molecules, such as aminoglycosides, marcolides, atalurens, etc, is its ability to target

select transcripts without disturbing global translation. By using specific guide RNAs, the authors direct a catalytically inactive dCas13 variant to regions downstream of the stop codon being investigated and perform a combination of readthrough assays to quantify SCR on exogenous reporter constructs as well as endogenous genes. They extend the efficacy of this system to induce SCR in premature termination codons in several other genes which can be crucial for developing therapeutics for genetic diseases caused by nonsense mutations. The authors attribute the SCR inducing potential of their system to transient ribosomal pausing caused by a molecular roadblock of gRNA-dCas13 complex.

> While the authors present compelling data to support their claim, many of their results lack control experiments that are crucial to validate the role of CRISPR/Cas13 in inducing SCR. There are several points that need to be addressed by the authors before acceptance.

I fully agree with this comment.

> Specific comments:

> 1) Page 5, Paragraph 1, last line: The authors mention the gRNAs target mRNAs, "proximally downstream (4-20 nucleotides) of the stop codon". However, the schematics in Fig 1A, 5A, EV2 and the gRNA sequence list (Materials and methods, Page 16) indicate otherwise. Do the authors mean 4-20 codons?

That is true

> 2) Page 6, line 18: The authors claim that the gRNA and dCas13 work in a transcript-specific manner to enhance SCR. To verify that both molecules act in concert, the authors need to provide control experiments (Fig 1B, 1C) where the cells are transfected with the luciferase reporter construct AGO1-UGA-3'UTR-Fluc and the AGO1 3'UTR targeting gRNA, but not the dCas13 construct. The same comment also applies to the following experiments indicated in Fig 4B, 5B, 6A and 6B

I also agree with this comment.

> 3) As an addition to point 2, it is essential to estimate the co-transfection efficiency of the reporter and the dCas13 constructs to uncouple the effects from gRNA alone and gRNA+dCas13 that might render variability in the readout. Additionally, the co-transfection efficiency of renilla luciferase used to normalize Fluc activity might also attribute to the variability in the readout. Having some representative values for absolute readouts of firefly as well as renilla luciferase in a supplementary table would be helpful.

I agree this is an important point.

> 4) Fig 1B, 1C: Electrophoretics-based RT-PCR cannot be used as a quantitative method to assess differences in expression of transfected genes as the observed signal saturates after multiple PCR amplification cycles. If the RT-PCR is done under limiting conditions where the amplification of cDNA of interest is relatively linear, then it should be mentioned in the materials and methods section. RT-qPCR analysis would be a more sensitive method to negate any effects that might stem from differences in expression levels of transfected genes. Alternately quantitative western blots can also be used for this. This comment also applies to Fig 2B, 5C, EV1A, EV2B.

The referee is correct. The authors must change that.

> 5) Page 7, Paragraph 1: How would the authors explain the fold differences in the enhancement of SCR by gRNA+dCas13 in MTCH2 (Fig EV2B) compared to AGO1 (Fig1B)

> ? It is also concerning that the relative luciferase activity of no-stop constructs for MTCH2 are so low in Fig EV2C.

I also agree with this comment.

> 6) Page 7, Paragraph 2: The western blot to detect Ago1x using anti-Ago1x, which is specific to the readthrough region of the protein, shows two bands that seem to increase upon induction by gRNA+dCas13 (Fig 2A,D) and three bands in Fig EV3.

Similar experiments performed with dCas13(Pgu) seem to give only one band. Can the authors comment on this discrepancy?

I agree

> 7) pcDNA mentioned in Fig EV3 need to be explained in the text or legend.

I agree, the authors can correct this point, I missed.

> 8) Page 8, Paragraph 1: Utilization of CRISPR-Cas9 system to target the AGO1-3'UTR (Fig EV3B) may not only downregulate the expression of the SCR isoform Ago1x, but also the major protein Ago1 by compromising the transcript stability. The authors would need to elaborate on how the 3'UTR was modified with CRISPR/Cas9 and present a control blot with anti-Ago1 to assert their claim.

This is also a point I missed. I think this is important to address it.

> 9) Page 9, Paragraph 1, 2nd last line: The authors report a 'significant' reduction in the ratio of the mean fluorescence intensity of mCherry to that of GFP, however, the graph (Fig 4A) shows only a modest reduction. The authors could change the verbiage.

I agree

>

> Minor:

> In general, there seems to be several inconsistencies with the brightness and contrast of western blot images throughout the

manuscript, which the authors can work on minimizing. There also seems to be minor formatting differences with regards to the alignment of figure titles, positioning of axes labels (eg: Fig 1C RT-PCR, Fluc), capitalization etc, which the authors can work upon.

I didn't notice inconsistencies with the images, but I would also recommend to follow the referee's advice on this point.

RESPONSES TO REVIEWERS' COMMENTS

We thank all the Reviewers for their critical analyses of our study and constructive suggestions. They have helped us immensely to improve the manuscript. We have addressed all the concerns of the Editor and the Reviewers.

Major changes to the manuscript:

1. We have included more results to rule out alternative explanations (cryptic promoter, alternative splicing, reinitiation, etc.) in reporter-based readthrough assays.
2. We have added more controls (gRNA only condition in all readthrough assays, a viral sequence as positive control in ribosome pausing assay, qRT-PCR to show equal mRNA, etc.) in our experimental demonstration of readthrough induction.
3. We have generated hereditary spherocytosis (the most common RBC membrane disorder) cellular model (nonsense mutation in *SPTA1* gene that encodes spectrin) to demonstrate the effectiveness of our strategy. We could achieve about 80% recovery of spectrin protein using CRISPR-dCas13 system.

Detailed point-by-point responses to their comments are given below. **Reviewers' comments are in bold and our responses are in italics.**

REFEREE #1

Manjunath et al described a CRISPR-based method to site-specifically promote premature termination codon (PTC) readthrough. They programmed dCas13 to bind to a mRNA region downstream to a PTC, thus presumably interfering with the translation termination process at the PTC by steric hindrance and in turn promoting readthrough. The approach may have therapeutic value for a broad range of genetic diseases caused by nonsense mutations. Data generally support the conclusions, and the manuscript is well written and presented. However, as a potential therapeutic approach, the reported technique can be strengthened by additional data.

Response: We thank the Reviewer for the appreciative words. We have revised the manuscript based on the suggestions given by all the Reviewers.

1. Are dCas13 and a readthrough agent (e.g., G418 or suppressor tRNA) additive or synergistic for promoting readthrough? Although such a readthrough agent tends to cause off-target readthrough at normal stop codons, a combinatorial approach may further enhance PTC readthrough.

Response: This is a very good suggestion. We have mentioned this possibility in the revised manuscript (Page 17, last paragraph). Since any readthrough-inducing agent should be used life-long, the concern of toxicity caused by indiscriminate induction of stop codon readthrough will be there even in combinatorial approach.

2. A dose-response experiment for dCas13 and gRNA to show which component may be limiting will be informative. This will inform on future optimization of the technique.

Response: dCas13 and gRNA form a functional complex with a stoichiometry of 1:1. Therefore, both will be required at the same level. We agree that optimization of in vivo doses of dCas13 and gRNA is required before attempting this as a therapeutic strategy.

3. I suggest the authors discuss on the delivery methods for their technique in a therapeutic setting, particularly in the case of in vivo delivery.

Response: As per this suggestion, we have discussed the challenge of in vivo delivery in the revised manuscript (Page 17, last paragraph).

REFEREE #2:

In this manuscript, the authors utilize the RNA targeting specificity of the CRISPR/Cas13 system and identify its novel application in elevating stop codon readthrough (SCR) in mammalian genes that naturally undergo SCR. What sets this system apart from other readthrough-inducing molecules, such as aminoglycosides, marcolides, atalurens, etc, is its ability to target select transcripts without disturbing global translation. By using specific guide

RNAs, the authors direct a catalytically inactive dCas13 variant to regions downstream of the stop codon being investigated and perform a combination of readthrough assays to quantify SCR on exogenous reporter constructs as well as endogenous genes. They extend the efficacy of this system to induce SCR in premature termination codons in several other genes which can be crucial for developing therapeutics for genetic diseases caused by nonsense mutations. The authors attribute the SCR inducing potential of their system to transient ribosomal pausing caused by a molecular roadblock of gRNA-dCas13 complex.

While the authors present compelling data to support their claim, many of their results lack control experiments that are crucial to validate the role of CRISPR/Cas13 in inducing SCR. There are several points that need to be addressed by the authors before acceptance.

Response: We thank the reviewer for the suggestions. We have performed all the suggested experiments with appropriate controls and addressed all the concerns of the Reviewer.

Specific comments:

1) Page 5, Paragraph 1, last line: The authors mention the gRNAs target mRNAs, "proximally downstream (4-20 nucleotides) of the stop codon". However, the schematics in Fig 1A, 5A, EV2 and the gRNA sequence list (Materials and methods, Page 16) indicate otherwise. Do the authors mean 4-20 codons?

Response: We thank the Reviewer for pointing out this mistake. We have changed that phrase to "4 to 40 nucleotides".

2) Page 6, line 18: The authors claim that the gRNA and dCas13 work in a transcript-specific manner to enhance SCR. To verify that both molecules act in concert, the authors need to provide control experiments (Fig 1B, 1C) where the cells are transfected with the luciferase reporter construct AGO1-UGA-3'UTR-Fluc and the AGO1 3'UTR targeting gRNA, but not the dCas13 construct. The same comment also applies to the following experiments indicated in Fig 4B, 5B, 6A and 6B

Response: We agree that expressing just gRNA without dCas13 is an important control experiment. As per this suggestion, we have included this control (vector control) in our experiments. Figures 1E, 2A, 3A, 4D, EV1D (AGO1-related), 5B (VEGFA-related), 6A and 6B (PTC-related).

3) As an addition to point 2, it is essential to estimate the co-transfection efficiency of the reporter and the dCas13 constructs to uncouple the effects from gRNA alone and gRNA+dCas13 that might render variability in the readout.

Response: This is a very good point. We performed the experiment as per this suggestion. We estimated the co-transfection efficiency using flow cytometry. The transfection efficiency of dCas13 and reporter constructs did not change significantly with gRNA (Fig EV2A).

Also, qRT-PCR analysis shows no significant change in the expression of reporters at mRNA level indicating no change in the transfection efficiency (Fig 1D, 4B, 5F, EV1E and EV4B).

Additionally, the co-transfection efficiency of renilla luciferase used to normalize Fluc activity might also attribute to the variability in the readout. Having some representative values for absolute readouts of firefly as well as renilla luciferase in a supplementary table would be helpful.

Response: We have provided the source data for all the experiments (including luciferase assays) performed.

4) Fig 1B, 1C: Electrophoretics-based RT-PCR cannot be used as a quantitative method to assess differences in expression of transfected genes as the observed signal saturates after multiple PCR amplification cycles. If the RT-PCR is done under limiting conditions where the amplification of cDNA of interest is relatively linear, then it should be mentioned in the materials and methods section. RT-qPCR analysis would be a more sensitive method to negate any effects that might stem from differences in expression levels of transfected genes. Alternately quantitative western blots can also be used for this. This comment also applies to Fig 2B, 5C, EV1A, EV2B.

Response: As per this suggestion, we have replaced all electrophoresis-based RT-PCR assays with quantitative Real-time-PCR data (Fig 1D, 1E, 2B, 4B, 4E, 5C, 5F, 6E, EV1E, EV1F, EV4B)

5) Page 7, Paragraph 1: How would the authors explain the fold differences in the enhancement of SCR by gRNA+dCas13 in MTCH2 (Fig EV2B) compared to AGO1 (Fig1B)?

Response: This is a very good point. The SCR efficiency depends on the context of the stop codon (PMID: 35570338). While MTCH2 shows ~59% SCR efficiency (as reported in PMID: 33028634), AGO1 shows ~20% SCR efficiency in reporter assays (as reported in PMID: 31330067). These observations suggest that MTCH2 stop codon is more susceptible to SCR than that of AGO1. This explains the difference in the enhancement of SCR by dCas13 in these two mRNAs.

Furthermore, the binding efficiencies of the AGO1 and MTCH2 gRNAs to their respective target mRNAs could vary. This can also influence the efficiency of SCR induction.

It is also concerning that the relative luciferase activity of no-stop constructs for MTCH2 are so low in Fig EV2C.

Response: We have replaced that data (now Fig EV4C).

6) Page 7, Paragraph 2: The western blot to detect Ago1x using anti-Ago1x, which is specific to the readthrough region of the protein, shows two bands that seem to increase upon induction by gRNA+dCas13 (Fig 2A,D) and three bands in Fig EV3. Similar experiments performed with dCas13(Pgu) seem to give only one band. Can the authors comment on this discrepancy?

Response: Ago proteins including Ago1 undergo post-translational Poly-ADP ribosylation induced by stress. The higher molecular weight bands we observe in some Ago1x Western blots may represent these isoforms (PMID: 21596313). We have mentioned this possibility with the appropriate citation in the revised manuscript (Page 9, first two lines).

7) pcDNA mentioned in Fig EV3 need to be explained in the text or legend.

Response: We have replaced 'pcDNA' with 'vector' as it was used as vector control.

8) Page 8, Paragraph 1: Utilization of CRISPR-Cas9 system to target the AGO1-3'UTR (Fig EV3B) may not only downregulate the expression of the SCR isoform Ago1x, but also the major protein Ago1 by compromising the transcript stability. The authors would need to elaborate on how the 3'UTR was modified with CRISPR/Cas9 and present a control blot with anti-Ago1 to assert their claim.

Response: The sequencing of the PCR-amplified proximal 3'UTR of AGO1 revealed 51 nucleotide deletion in that region (Appendix Figure S3). The procedure is elaborated in the "Materials and Methods" under the subsection "CRISPR-Cas9 mediated deletion". We have provided the control blot with anti-Ago1 antibody to support our claim (Fig EV4 B).

Nevertheless, Ago1 levels will not affect our conclusions as this assay was performed to show the specificity of the anti-Ago1x antibody used in the study.

9) Page 9, Paragraph 1, 2nd last line: The authors report a 'significant' reduction in the ratio of the mean fluorescence intensity of mCherry to that of GFP, however, the graph (Fig 4A) shows only a modest reduction. The authors could change the verbiage.

Response: As per this suggestion, we have changed that statement to 'modest' reduction (Page 12, first paragraph). Note that this is the difference expected from a transient ribosomal pausing event. Differences larger than this (observed in our positive control) indicate ribosomal stalling, which can have other consequences including ribosomal disassociation (PMID: 28715909).

Minor:

In general, there seems to be several inconsistencies with the brightness and contrast of western blot images throughout the manuscript, which the authors can work on minimizing.

Response: We have not made any post-capture changes (including brightness and contrast) to the Western blot images. The differences seen across the Western blot images could be due to the differences in the exposure time during image acquisition. We have provided the uncropped source data for all Western blot images.

There also seems to be minor formatting differences with regards to the alignment of figure titles, positioning of axes labels (eg: Fig 1C RT-PCR, Fluc), capitalization etc, which the authors can work upon.

Response: We thank the Reviewer for pointing these issues. We have taken care of them in the revised manuscript.

REFEREE #3

The manuscript from Manjunath LE et al. aims at using dCAS13 to stimulate stop codon readthrough in an mRNA-specific way. The general idea is to target a downstream sequence of the stop codon to create a roadblock that would stall the ribosome at the stop codon. The idea is interesting although not completely original because several proteins have been already shown to play such role.

Response: We agree that some endogenous trans-acting factors, including proteins, have been shown to induce SCR (Example, PMIDs: 24949972, 37714299, 31330067). We have provided this information in detail with references in the 3rd paragraph of the Introduction. In this study, we have demonstrated that the SCR induction can be achieved by exogenously introduced dCas13-gRNA complex.

The authors also explain this could be very interesting for stimulating PTC readthrough for therapeutics purposes. They are right to mention that currently there is no mRNA-specific way to stimulate readthrough, and that all current methods stimulate readthrough in a global way. However, I do not understand the rationale of the approach from a therapeutic point of view. Indeed, this approach would require introducing into the cells of the patients the dCAS13, and the gRNA. In this case, why not directly transform the cells with the WT mRNA corresponding to the defective gene, to restore a complete expression of the protein? This seems to me much more direct and efficient than trying to stimulate the readthrough and also probably subject to fewer harmful side effects.

Response: The reviewer has raised an important and valid question. Yes, expressing the healthy version of the mRNA seems like a better strategy than expressing dCas13-gRNA complex. However, this is not true in all conditions. In the conditions

explained below, induction of SCR across the PTCs is a better option than expressing the healthy version of the mutant mRNA.

- 1. In some conditions the defective gene/mRNA is very long. For example, the coding sequence in the human DMD transcript (NM_004006.3) is 11058 nucleotides long. Mutations in this gene cause Duchenne Muscular Dystrophy. It will be a challenge to deliver and express such a big transcript in vivo.*
- 2. Sometimes truncated proteins generated due to PTCs can have dominant negative effects, which contribute to the pathology of the disease. For example, SOX10 mutations in Waardenburg-Shah syndrome (PMID: 11546831). In such conditions expressing the healthy version of the protein might not provide therapeutic benefits. However, induction of SCR will reduce truncated proteins, and therefore pathologies associated with it.*
- 3. Overexpression of a protein in some cases can have pathological consequences due to gain of function (e.g., MeCP2 duplication syndrome). In case of Rett's syndrome (caused due to mutations in MeCP2), even mild overexpression of MeCP2 can lead to progressive neurological disorder (PMID: 15351775). Therefore, in such conditions, it is difficult to express an mRNA without risking the overexpression and associated adverse effects.*
- 4. Furthermore, it is difficult to determine the optimal level of expression as the expression of a gene varies across cell types and tissue types. This problem will be minimized if we employ induction of SCR as it acts on already expressed mRNAs as per the needs of the cells and the tissues.*

Thus, at least in the conditions described above, induction of SCR across PTCs seems to be a better strategy than expressing the healthy version of the mRNA itself. Our study demonstrates a transcript-selective strategy based on dCas13-gRNA to achieve SCR induction. We have included all these points in our discussion (penultimate paragraph). Thanks to this comment, the discussion part has immensely improved.

All the work and reasoning are based on the use of these reporter systems, so it is essential that all the controls are correctly performed to eliminate alternative explanations.

Response: In addition to reporter assays, we provide Western blot evidence to support our claim on the successful dCas13-gRNA complex-induced readthrough. The molecular weight of the protein bands in all these experiments are consistent with the readthrough products. (Fig.1C, 2, and EV1D (AGO1); Fig. 5B (VEGFA); Fig. 6 (GFP, HBB, TP53 and SPTA1).

Furthermore, as suggested by the Reviewer, we have now included additional controls in our SCR induction experiments, luminescence-based ribosome pausing assay and RiboPuromycylation assay (See Figures 1E, 2A, 3A, 4D, EV1D (AGO1-related), 5 (VEGFA related), 6 (PTC related). As described below in detail we have eliminated alternative explanations.

Actually, It is now well established that reporter systems suffer from many potential biases (alternative promoter, alternative splicing, leaky scanning, reinitiation...). They must be rigorously considered and excluded to support the working hypotheses; I am afraid this is not the case on this manuscript.

Response: We agree that the above-mentioned phenomena can potentially confound reporter-based assays. Evidence against these alternative possibilities are provided in the manuscript and explained below:

- 1. We have demonstrated the induction of SCR by dCas13-gRNA in endogenous AGO1 (Fig 2A, 2C), VEGFA (Fig 5B), TP53 (Fig 6C) and SPTA (Fig 6D, newly added data) mRNAs by Western blotting. In addition, we have performed Western blotting to demonstrate SCR induction in an exogenous reporter of AGO1 (Fig 1C and EV1D, newly added data). In all these experiments, the molecular weight of the protein band obtained correspond to that of the SCR product. These observations rule out alternative promoter, translation reinitiation caused by leaky scanning, and alternative splicing, which would have resulted in shorter products than what we have observed.*
- 2. We have demonstrated dCas13-gRNA mediated induction of SCR using in vitro transcribed RNA and rabbit reticulocyte (lacks the nucleus) lysate system (Fig 1E). This rules out alternate promoter activity and alternative splicing events.*
- 3. We have sequenced the cDNAs of the mRNAs generated from the reporter construct in cells transfected with dCas13-gRNA complex. These results also*

do not support alternative splicing (Fig EV2B, EV2C, Appendix Fig S1 and S2).

4. Results from our previous studies (PMIDs 24949972, 31330067, 33028634) also rule out these alternative phenomena.
5. The insertion between the coding sequence of AGO1 and the reporter is just 99 nucleotides (not > 600 nucleotides as mentioned below by the Reviewer. We deeply regret this confusion) without an in-frame start codon. This also makes the alternative possibilities unlikely.

These points are discussed in the revised manuscript (Results, Page 8).

It is curious to choose two genes for which the readthrough is programmed (Ago1 and VEGFA) to demonstrate that it is possible to stimulate it using dCas13. On Ago1 the authors have previously shown that there was a stimulation of readthrough efficiency by the small RNA let7-A which seems to bind to the region targeted by dCas13. The authors do not even discuss this possible competition and its consequences.

Response: We thank the Reviewer for pointing this out. In the revised manuscript we have discussed this aspect (Page 11, second paragraph). The SCR of AGO1 is positively regulated by let-7a micro-RNA. Both let-7a micro-RNA and the dCas13-gRNA target same region in the 3'UTR of AGO1 mRNA. Our results show that, like let-7a, exogenous dCas13-gRNA can induce SCR of AGO1. Since the gRNA is 28-30 nucleotides long with 100% complementarity with AGO1 mRNA, dCas13-gRNA is likely to be better inducer than let-7a micro-RNA, which has a 11 nucleotide stretch of partial complementarity with the AGO1 mRNA (PMID: 31330067).

It is technically very difficult to compare the efficiency of let-7a micro-RNA and the gRNA-dCas13 complex in inducing SCR. The former requires expression of just the microRNA, and the latter requires expression of both gRNA and dCas13. We cannot be sure that both systems are present at the same (or even comparable) levels inside the cell for comparison.

If the aim of the work is a therapeutic approach, why not have focused the approach on PTCs present in genetic diseases as at the end of the

manuscript? for which it is unfortunately not clear if the induction efficiency will have a real impact or not.

Response: Development of a potential therapeutic approach is one application of our strategy based on dCas13. In addition, this approach can also be used to modulate natural SCR events and can serve as a tool to understand the functional significance of SCR events. This is important because, other tools such as siRNA cannot be used to understand the function of endogenous SCR events as siRNAs bring down the mRNA levels resulting in downregulation of both the canonical isoform and the SCR product. We have made this clear in the discussion (Last paragraph of Results and Discussion).

As per this suggestion, we generated a cellular model of PTC-mediated hereditary spherocytosis, the most common RBC membrane disorder, using K562 cells (human erythroleukemia cells) with a patient-specific mutation in SPTA1 gene that encodes alpha-spectrin. We have demonstrated dCas13-mediated induction of SCR and generation of full length alpha spectrin in these cells. In these mutant cells, we could achieve spectrin protein levels about 80% of that in parental wild-type cells (Fig 6D). It should be noted that even partial restoration of functional proteins (e.g., 20-30% of normal dystrophin levels in DMD patients) can provide therapeutic benefits in many genetic diseases (PMID: 24773318).

I have several important comments for the authors:

-P6, L10 (figure 1B). a western-blot is necessary to confirm readthrough by visualising a fusion protein at the correct size. Currently the only conclusion is that the spacer region promotes Fluc activity.

Response: We have performed the suggested experiment and show the fusion protein (with the FLAG-HA tag) of correct size corresponding to SCR product (Fig 1C and EV1D).

Many alternative mechanisms could explain the skipping of the stop codon (i.e alternative splicing removing the UAA codon, re-initiation downstream the UAA) with a so long insertion (>600nt) these events are likely to occur. A nanopore sequencing will be reassuring about the integrity of the mRNA and the absence of alternative promoters/initiators.

Response: We believe there is a misunderstanding here. We deeply regret this confusion. The insertion the Reviewer is referring to is not >600 nucleotides. It is just 99 nucleotides of the proximal 3'UTR of AGO1 (mentioned in the legend and in the Results) that is necessary and sufficient to cause SCR in AGO1 mRNA.

We would like to clarify the lengths of the different parts of the construct to avoid confusion:

Partial coding sequence of AGO1 – 696 nucleotides (distal end)

Proximal 3'UTR of AGO1 – 99 nucleotides (does not have AUG)

Linker sequence – 30 nucleotides (does not have AUG)

Firefly luciferase – 1650 nucleotides (does not have AUG in the beginning)

As explained above (page 10-11) in 5 points, we have ruled out alternative promoter activity by performing Western blotting (Fig 1C and EV1D), in vitro translation of reporter RNAs generated by in vitro transcription (Fig 1E) and by sequencing the reporter transcripts (Fig EV2B, EV2C, Appendix Fig S1 and S2).

Fig 1A is misleading because the grey region of ago1 is more than 600nt long, F-Luc is around 1kb, so please be proportional in the schematic.

Response: We again regret the confusion here. The grey region is 99 nucleotide, not > 600 nucleotides (see the image shown above). Nonetheless, we have altered the schematic as per this suggestion and tried to make it as proportional as possible.

Sequences would also be necessary in supplementary data to clearly understand which parts are present and removed between panels B and C figure 1. because this is certainly not only the blue part that is missing in panel C

Response: We have included the sequences in Appendix Fig S2.

In the revised manuscript these Figures are numbered Fig EV1E and F (Previously 1B and 1C). Only difference between panel EV1E and F constructs is that the stop codon after AGO1 coding sequence has been removed in panel EV1F.

page 7, Figure 2A, 2B. Why anti-ago1x reveals two bands in panel A? this is not mentioned at all neither in the text nor in the legend of the figure.

Response: Ago proteins including Ago1 undergo post-translational Poly-ADP ribosylation induced by stress. The higher molecular weight bands we observe in Ago1x Western blots may represent these isoforms (PMID: 21596313). We have mentioned this in the revised text (Results, page 9).

Why the readthrough form is not detected by anti-ago1 ?

Response: The molecular weight difference between Ago1 and Ago1x is just ~ 3 kDa. This difference cannot be resolved in the electrophoretic conditions we have used. This is explained in our previous paper (PMID: 31330067).

at this stage, all changes in luciferase activities or protein level could be explain by alternative explanations.

Response: As explained above (page 10-11) in 5 points, we have ruled out alternative explanations by performing Western blotting (Fig 1C and EV1D), in vitro translation of reporter RNAs generated by in vitro transcription (Fig 1E) and by sequencing the reporter transcripts (Appendix Fig S1 and S2).

There is no direct evidence that dCas13 directly stimulates readthrough.

Response: To address this, we have performed in vitro translation and used extracts from cells transfected with dCas13-gRNA to demonstrate induction of SCR (Fig 1E).

-P8, L3, there is no demonstration that the effect on translation observed come from the higher synthesis of Ago1x (the readthrough form of ago1). To demonstrate the link they should use a siRNA against Ago1 to prove the direct link.

Response: Ago1 participates in miRNA-mediated gene silencing. Therefore, knockdown of AGO1 alone (irrespective of SCR levels) can enhance the global translation. We have demonstrated this by knocking down AGO1 using Cas13

(catalytically active form) (Fig EV3A and EV3C). We have also overexpressed Ago1x and shown that it can enhance global translation (Fig EV3D and EV3E), which is consistent with dCas13 mediated induction results (Fig 3A and 3B).

Moreover there are more quantitative way to quantify readthrough (polysome fractions or click-it kits coupled with alexa fluorophores) rather than ponceau staining and total puromycin staining.

Response: As per this suggestion we have performed click-it based assay to quantify global translation (Fig 3B). The results are in agreement with RiboPuromylation assays.

P9, figure 4. the effects observed are very marginal, if any and can hardly explain the previously described translation defects (0.41 vs 0.37 probably)

Response: We agree that the differences are marginal, however this is the difference expected from a transient ribosomal pausing event. Differences larger than this (observed in our positive control) indicate ribosomal stalling, which can have other consequences including ribosomal disassociation (PMID: 28715909).

Did the authors quantify mRNA level of the reporter? because ribosome stalling is known to induce mRNA degradation through the RQC pathway.

Response: Yes, we have quantified the mRNA levels, and the data is included in the revised manuscript (Fig 4B). We did not observe any significant change in the levels of the mRNA.

Moreover P2A sequences themselves induce a strong ribosome pausing (and drop off) (PMID: 25621616 + 8112307 + 28526819) this is far from a good idea to include them in a reporter system dedicated to quantify ribosomal pausing.

Response: The reviewer has raised a valid question. We would like to make following arguments in favour of our approach:

- 1. This dual-fluorescence reporter construct with P2A sequence has been previously used to investigate ribosomal stalling (PMID: 28065601; 31768042).*
- 2. We have used positive (K20) and negative (K0) controls to show that the assay indeed captures ribosome stalling.*

3. *There is no difference in the sequence (i.e., GFP-P2A-AGO1-3'UTR-P2A-mCherry) in our control and test experiments (Fig 4A). Hence, even if there is pausing due to the inherent property of the sequence itself, it will be there in both the conditions. We clearly observe a marginal decrease in mCherry/GFP fluorescence ratio consistent with ribosomal pausing due to dCas13-gRNA complex (Fig 4A).*
4. *Furthermore, we have confirmed dCas13-gRNA complex-induced ribosomal pausing by another assay based on luminescence (Fig 4D).*

P10- Readthrough is probably less than 50% efficient in presence of dcas13 (although this is never properly quantified in the manuscript), especially by mixing RRL and HeLa cells extracts. So variations of Fluc activity due to RT is marginal in comparison to the luc activity generated from ribosomes that stopped at the UAA codon. Does mRNA level has been check in RRL after the experiment to be sure the presence of gRNA of dCAS13 does not induced luc mRNA degradation?

Response: We agree with the Reviewer that mRNA level should be checked. And, we have provided the data in Fig 1E and 4E. Extracts containing dCas13 and gRNA didn't induce mRNA degradation.

The Reviewer has also raised a query on quantification of induction of readthrough. We would like to summarize the quantification shown in all the figures:

- *AGO1: About 2.5-fold induction based on Western blotting results of endogenous Ago1x (Fig 2A)*
- *VEGFA: About 2.5-fold induction based on Western blotting results of endogenous VEGFAx (Fig 5B)*
- *MTCH2: About 2-fold induction based on luciferase-based reporter assay (Fig EV3B)*
- *SPTA1: About 80% of the parental wild-type cells (Fig 6D).*
- *TP53: Sufficient enough to show downstream effects (increase in the expression of CDKN1A (P21) and BAX mRNAs) (Fig EV5)*

P10, Fig 4B. Before concluding that they have demonstrated ribosome pauses, they should demonstrate that their system is capable of detecting pauses

when bona-fide readthrough inducers are used. Controls are missing in this experiments.

Response: As per this suggestion we have included the following controls:

- 1. A Sequence from Venezuelan Equine Encephalitis Virus (VEEV) that is reported to have a stem-loop structure and known to induce SCR (PMID: 32560154, 21525127) is included as positive control, and it shows expected delay in the appearance of luminescence (Fig 4C).*
- 2. The proximal 3'UTR of AGO1, which itself can drive SCR, also shows delay in the appearance of luminescence. (Fig 4C)*
- 3. gRNA only control (without dCas13) is included, and it does not show delay in the appearance of luminescence. (Fig 4D)*

Moreover, how can we deal with the previous model published by the authors indicating that Ago1x self-regulate its own readthrough using let7-a miRNA? Why this is not shown? is there any competition between let7-a complex and dcas13?

Response: The SCR of AGO1 is positively regulated by let-7a micro-RNA. Both let-7a micro-RNA and the dCas13-gRNA target the same region in the 3'UTR of AGO1 mRNA. Our results show that, like let-7a, exogenous dCas13-gRNA can induce SCR of AGO1. Since the gRNA is 28-30 nucleotides long with 100% complementarity with AGO1 mRNA, dCas13-gRNA is likely to be better inducer than let-7a micro-RNA, which has a 11 nucleotide stretch of partial complementarity with the AGO1 mRNA (PMID: 31330067). In the revised manuscript we have discussed this aspect (Page 11).

It is technically very difficult to compare the efficiency of let-7a micro-RNA and the gRNA-dCas13 complex in inducing SCR. The former requires expression of just the microRNA, and the latter requires expression of both gRNA (RNA) and dCas13 (protein). We cannot be sure that both systems are present at same (or even comparable) levels inside the cell for comparison.

P10- fig 5. Proper loading controls must be realized, Ponceau is not enough proportional to be relevant for quantification

Response: The samples are conditioned medium from transfected cells. They contain only secreted proteins. Since there are no established secreted proteins that

can serve as loading control, we have used Coomassie staining to quantify total protein in the sample.

P11 Fig5F. why not using gold standard dual reporter to quantify properly readthrough?

Response: As shown previously VEGFA does not show SCR in dual reporter assay (PMID: 28442579). This is because, the coding sequence of VEGFA is required for the SCR along with its proximal 3'UTR. There is an interaction between the proximal 3'UTR and the proximal coding sequence of VEGFA (doi.org/10.21203/rs.3.rs-186157/v1). This interaction is not possible in dual reporter assays.

P12. Fig6. Some controls are also missing here. For example, there is no mRNA quantification. What about the possibility that dcas13 stabilize the mRNA? then background level of readthrough will be sufficient to see the protein by WB.

Response: We have provided the qRT-PCR data for the RNA levels; We don't observe any significant increase (Fig 6E). Thus, background levels of SCR does not explain our results.

P13 - Fig 6 do both panels of fig6C come from the same gel?

Response: The results shown are from the same samples run in two separate gels.

G418 is used at which concentration?

Response: G418 was used at a concentration of 2 mg/ml. We have given this information in the Figure legend.

Dear Prof. Eswarappa,

Thank you for the submission of your revised manuscript. We have now received the enclosed report from referee 3, who also assessed your response to referee 2's comments. I am happy to say that referee 3 supports the publication of your work now and we can therefore in principle accept it for publication.

Only some more minor editorial requests still need to be addressed :

- Please separate the results and discussion sections, as your ms has 6 main figures and will therefore be published as a full article.
- Please correct the conflict of interest subheading to "Disclosure Statement and Competing Interests"
- Please move the references to after the "Disclosure and Competing Interests" statement.
- Please remove the authors credits from the ms file. All credits need to be entered during online ms submission.
- Please submit with your final ms a completed author checklist that can be found here: <<https://www.embopress.org/page/journal/14693178/authorguide>>. The completed list will also be part of our transparent peer-review process file.
- Please enter all funding info also during online ms submission, some info is currently missing in this system but is present in the ms file.
- The APPENDIX FILE does need a title page with table of content and page numbers.
- At the final image size, the text in the synopsis image is slightly blurred. If possible, please send us a new image with better text at the exact size of 550 pixels wide.
- Please also submit with your final ms A) a short (1-2 sentences) summary of the findings and their significance, B) 2-3 bullet points highlighting key results.
- Please upload the source data (SD) for main figures as one folder per figure, while the folders for all EV figures can be grouped and uploaded as one zipped folder.
- Please address these comments by our data editors:
 1. Please note that a separate 'Data Information' section is required in the legends of figures 1c-e; 2a-c; 4a-b; 5b-d, f-g; 6a-d; EV 1b, e-f; EV 4b-c.
 2. Please note that information related to n is missing in the legend of figure 4e.
 3. Please note that the error bars are not defined in the legend of figure 4e.

I would like to suggest some changes to the abstract that needs to be written in present tense. Please let me know whether you agree with the following:

Stop codon readthrough (SCR) is the process where translation continues beyond a stop codon on an mRNA. Here, we describe a strategy to enhance or induce SCR in a transcript-selective manner using a CRISPR-dCas13 system. Using specific guide RNAs, we target dCas13 to the region downstream of canonical stop codons of mammalian AGO1 and VEGFA mRNAs, known to exhibit natural SCR. Readthrough assays reveal enhanced SCR of these mRNAs (both exogenous and endogenous) caused by the dCas13-gRNA complexes. This effect is associated with ribosomal pausing, which has been reported for several SCR events. Our data show that CRISPR-dCas13 can also induce SCR across premature termination codons (PTCs) in the mRNAs of green fluorescent protein and TP53. We demonstrate the utility of this strategy in the induction of readthrough across the thalassemia-causing PTC in HBB mRNA and hereditary spherocytosis-causing PTC in SPTA1 mRNA. Thus, CRISPR-dCas13 can be programmed to enhance or induce SCR in a transcript-selective and stop codon-specific manner.

Referee #3:

As requested, I also assessed comments to ref 2 (I was referee 3).

In a general point of view, the authors have addressed most of our concerns and provided convincing responses to our comments. Their new experiments contribute to a more persuasive story, which is very interesting. I think this is a very clear demonstration of the effect of the dCAS13 on stop codon readthrough stimulation.

While I concur with the authors that in certain specific instances, the dCAS13 approach may be preferable to expressing the entire mRNA, I believe that in the majority of cases, a full mRNA replacement would be more effective. Therefore, I am not entirely convinced of its therapeutic relevance, although the work is interesting in by itself.

All editorial and formatting issues were resolved by the authors.

Prof. Sandeep Eswarappa
Indian Institute of Science
C V Raman
Bengaluru, Karnataka 560012
India

Dear Prof. Eswarappa,

I am very pleased to accept your manuscript for publication in the next available issue of EMBO reports. Thank you for your contribution to our journal.

Yours sincerely,
